

# Variability in oxygen isotopic fractionation of enzymatic O₂ consumption

Carolina F. M. de Carvalho[1], Moritz F. Lehmann[1], Sarah G. Pati[1,2]

[1] Department of Environmental Sciences, University of Basel, 4056 Basel, Switzerland

[2] Department of Environmental Geosciences, Centre for Microbiology and Environmental Systems Science, University of Vienna, 1090 Vienna, Austria

*Correspondence to*: Sarah G. Pati (sarah.pati@univie.ac.at)

**Abstract.** Stable isotope analysis of $O_2$ has emerged as a valuable tool to study $O_2$ dynamics at various environmental scales, from molecular mechanisms to ecosystem processes. Despite its utility, there is a lack of fundamental understanding of the

large variability observed in $O_2$ isotopic fractionation at the environment- and even enzyme-level. To expand our knowledge on the potential causes of this variability, we determined $^{18}O$-kinetic isotope effects (KIEs) across a broad range of $O_2$-consuming enzymes. The studied enzymes included nine flavin-dependent, five copper-dependent, and one copper-heme-dependent oxidases, as well as one flavin-dependent monooxygenase. For twelve of these enzymes, $^{18}O$-KIEs were determined for the first time. The comparison of $^{18}O$-KIEs, determined in this and previous studies, to calculated $^{18}O$-equilibrium isotope

effects revealed distinct patterns of O-isotopic fractionation within and between enzyme groups, reflecting differences in active-site structures and $O_2$-reduction mechanisms. Flavin-dependent $O_2$-consuming enzymes exhibited two distinct ranges of $^{18}O$-KIEs (from 1.020 to 1.034 and from 1.046 to 1.058), likely associated with the rate-limiting steps of two different $O_2$-reduction mechanisms (sequential vs. concomitant 2-electron transfer). In comparison, iron- and copper-dependent enzymes displayed a narrower range of $^{18}O$-KIEs, with overall lower values (from 1.009 to 1.028), which increased with the degree of

$O_2$ reduction during the rate-limiting step. Similar to flavin-dependent $O_2$-consuming enzymes, copper-dependent $O_2$-consuming enzymes also featured two main, yet narrower, ranges of $^{18}O$-KIEs (from 1.009 to 1.010 and from 1.017 to 1.022), likely associated with the rate-limiting formation of a copper-superoxo or copper-hydroperoxo intermediate. Overall, our findings support generalizations regarding expected $^{18}O$-KIEs ranges imparted by $O_2$-consuming enzymes and have the potential to help interpret stable $O_2$ isotopic fractionation patterns across different environmental scales.

**1 Introduction**

Stable isotope analysis of $O_2$ has proven to be a valuable tool for tracking and quantifying environmentally relevant $O_2$ dynamics across different spatial and temporal scales. On a large environmental scale, stable isotope analysis of $O_2$ has been most commonly used in aquatic studies to estimate the productivity of oceans and lakes (Luz and Barkan 2000; Hendricks et al. 2005; Gammons et al. 2014; Bocaniov et al. 2015; Bogard et al. 2017), but also as a tracer of ocean circulation (Kroopnick

and Craig 1976; Bender 1990; Levine et al. 2009), and to estimate historical changes in the global hydrological and $O_2$ cycle



(Petit et al. 1999; Severinghaus et al. 2009; Blunier et al. 2012). On a smaller environmental scale, it has been used to study the dynamics of $O_2$ consumption by plants (Guy et al. 1992, 1993; Ribas-Carbo et al. 1995; Helman et al. 2005), microorganisms (Helman et al. 2005; Stolper et al. 2018; Ash et al. 2020), and humans (Epstein and Zeiri 1988; Zanconato et al. 1992).

In most environmental applications of $O_2$ isotope analysis, biological $O_2$ consumption is the main process driving and modulating the changes in the $^{18}O/^{16}O$ and $^{17}O/^{16}O$ ratios of $O_2$. Spatial and/or temporal changes in $O_2$ isotope ratios are referred to as isotopic fractionation and can be quantified with, for example, $^{18}\varepsilon$ values (see Eq. (1)) (Coplen 2011):

$$^{18}\varepsilon = \ln\left(\frac{(^{18}O/^{16}O)}{(^{18}O/^{16}O)_0}\right)/\ln\left(\frac{[O_2]}{[O_2]_0}\right) \tag{1}$$

Here, $(^{18}O/^{16}O)$ and $(^{18}O/^{16}O)_0$ represent the isotopic ratios of $O_2$ in a sample at a given timepoint, and in a reference sample
(typically reflecting initial conditions or original source), respectively, and $[O_2]/[O_2]_0$ represents the fraction of $O_2$ remaining after partial consumption. Typically, $^{18}\varepsilon$ values are indicative of a specific reactive process and may thus be used to identify, track, and quantify $O_2$ consumption processes in the environment. However, the magnitude of $^{18}\varepsilon$ values measured for bulk biological $O_2$ consumption, considered to be predominantly respiration, varies considerably. Specifically, in aquatic environments, $^{18}\varepsilon$ values determined for respiratory $O_2$ consumption range from -7 ‰ to -26 ‰ (Kiddon et al. 1993; Helman
et al. 2005; Wang et al. 2008; Levine et al. 2009; Bocaniov et al. 2015). Although it has been suggested that the observed variability in $^{18}\varepsilon$ values can be explained by the different types of organisms consuming $O_2$, the availability of light (e.g. effect of photosynthesis and/or photoinhibition pathways) and the main metabolic pathway (Mader et al. 2017), there is still no fundamental understanding of the underlying causes of this variability. The uncertainty associated with the O-isotopic fractionation of respiratory $O_2$ consumption has substantial implications for the application of $O_2$ isotope analysis to study
ecosystem respiration on an environmental scale. For instance, most $O_2$-isotope applications to study aquatic ecosystems require assuming a constant $^{18}\varepsilon$ value for respiration to estimate respiration rates (Wang et al. 2008; Bocaniov et al. 2012; Bogard et al. 2017). Consequently, these respiration rates are prone to considerable error depending on the accuracy of chosen community respiration $^{18}\varepsilon$ value (Hotchkiss and Hall 2014).

     To improve the quantification of gross $O_2$ production in aquatic environments, an increasing number of studies are
applying the triple oxygen isotope (TOI) method (Luz and Barkan 2000; Hendricks et al. 2005; Juranek and Quay 2013; Jurikova et al. 2016). In TOI applications, changes in $^{17}O/^{16}O$ ratios relative to changes in $^{18}O/^{16}O$ ratios along $O_2$ concentration gradients are quantified as $\lambda$ values (Miller 2002; Sharp et al. 2018). $\lambda$ values for biological $O_2$ consumption range between 0.51 and 0.53 (Young et al. 2002; Luz and Barkan 2005; Ash et al. 2020; Hayles and Killingsworth 2022), with a value of 0.518 typically assumed for marine respiration (Luz and Barkan 2009; Juranek and Quay 2013). Because $\lambda$ values vary less
than $^{18}\varepsilon$ values for respiration, TOI analysis often improves gross $O_2$ production estimates. However, the overall robustness of $\lambda$ values representative for respiration, and other biological $O_2$-consuming processes, has been recently questioned in other studies (Stolper et al. 2018; Ash et al. 2020; Sutherland et al. 2022a, 2022b).





In addition to environmental applications, stable isotope analysis of $O_2$ has also been applied on a molecular scale to uncover reaction mechanisms of substrate oxidation and $O_2$ reduction by $O_2$-consuming enzymes (Roth and Klinman 2005).

Specifically, oxygen equilibrium isotope effects ([18]O-EIEs) and oxygen kinetic isotope effects ([18]O-KIEs) are used as mechanistic probes to assess the rate-limiting steps in $O_2$-consuming enzymatic reactions (Roth and Klinman 2005). [18]O-EIEs can be calculated or experimentally determined for the reversible formation of free, or ligand-bound, reactive oxygen species (Roth and Klinman 2005; Lanci et al. 2007; Mirica et al. 2008), such as superoxide ($O_2^{\bullet-}$, see Eq. (2)), and reflect the ratio of reaction rate constants of light ($^{16}O^{16}O$) versus heavy ($^{18}O^{16}O$) isotopologues of $O_2$, as shown in Eq. (3).

$$O_2 \underset{k_r}{\overset{k_f}{\leftrightharpoons}} O_2^{\bullet-} \tag{2}$$

$$^{18}O\text{-}EIE = \frac{^{18}O\text{-}KIE_f}{^{18}O\text{-}KIE_r} = \frac{^{16}k_f/^{18}k_f}{^{16}k_r/^{18}k_r} \tag{3}$$

Where $k_f$ and $^{18}O\text{-}KIE_f$ are the reaction rate constant and KIE of the forward reaction between $O_2$ and $O_2^{\bullet-}$, $k_r$ and $^{18}O\text{-}KIE_r$ are the reaction rate constant and KIE of the reverse reaction, and $^{16}k$ and $^{18}k$ denote reaction rate constants for the light and heavy isotopologues of $O_2$, respectively. Experimentally determined [18]O-KIEs reflect the O-isotopic fractionation occurring in all

elementary reaction steps up to, and including, the rate-limiting step (Roth and Klinman 2005). Experimental [18]O-KIEs are thus often referred to as observable or apparent [18]O-KIEs, and they reflect an averaged O-isotope effect for both O atoms in $O_2$. Apparent [18]O-KIEs are related to $^{18}\varepsilon$ values as shown in Eq. (4).

$$^{18}O\text{-}KIE = (^{18}\varepsilon + 1)^{-1} \tag{4}$$

Typically, apparent [18]O-KIEs closely reflect the intrinsic [18]O-KIE of the rate-limiting step, which can be the binding of $O_2$ to

the active site, or an elementary $O_2$ reduction step (Roth and Klinman 2005). Because intrinsic [18]O-KIEs cannot be easily calculated, [18]O-EIEs are often used as a reference to assign experimentally determined [18]O-KIEs to a specific rate-limiting step (Roth and Klinman 2005). Together, these parameters can help to elucidate the intermediate species, and the number of electrons and protons transferred to $O_2$, before and during the rate-limiting step (Roth and Klinman 2003; Mirica et al. 2008; Humphreys et al. 2009).

All biological $O_2$ consumption, including respiration, detoxification, and biosynthesis, is ultimately carried out by $O_2$-consuming enzymes. Therefore, the variability in the isotopic fractionation of $O_2$ observed at both small and large environmental scales may be initially attributed to that observed at the enzyme level. However, few attempts have been made to relate $O_2$ isotopic fractionation occurring at the enzyme level to that occurring at larger environmental scales (Guy et al. 1987, 1989). So far, approximately 850 $O_2$-consuming enzymes have been described by *The Nomenclature Committee of the*

*International Union of Biochemistry and Molecular Biology database* (McDonald et al. 2009). Yet, comparatively few have





been comprehensively studied. $O_2$-consuming enzymes have evolved specialized active-site structures to overcome the kinetic limitations of $O_2$ reduction and to exploit the reactivity of the reduced oxygen species for productive redox catalysis (Malmstrom 1982; Klinman 2007; Frey and Hegeman 2007). These active site structures are typically flavin-, copper- or iron-dependent structures that, via the formation of radical intermediates with organic cofactors, or interactions with transition-state

metals, can rapidly and easily reduce $O_2$ (Malmstrom 1982; Bugg 2001; Bento et al. 2006; Frey and Hegeman 2007; Pimviriyakul and Chaiyen 2020). There are two major groups of $O_2$-consuming enzymes: oxidases and oxygenases. Oxidases catalyze the transfer of one-, two-, or four-electrons from their substrate(s) to $O_2$, reducing $O_2$ to either hydrogen peroxide ($H_2O_2$) or water ($H_2O$) (Malmstrom, 1982). The transfer of electrons from a given substrate to $O_2$ typically occurs in two separate steps through oxidation and reduction of the enzyme. Substrate oxidation by the oxidized enzyme occurs in the

reductive half-reaction, and $O_2$ reduction by the reduced enzyme occurs in the oxidative half-reaction. Oxidases are more often involved in catabolic processes, oxidizing substrates like alcohols, amines, and amino acids (Medda et al. 1995; Finney et al. 2014; Pimviriyakul and Chaiyen 2020). For example, glucose oxidase, one of the most well studied oxidases, catalyzes the oxidation of β-D-glucose to D-glucono-δ-lactone and $H_2O_2$. This reaction is part of the catabolic process that breaks down glucose, providing energy and components needed for anabolic reactions (Bauer et al. 2022). Oxygenases, on the other hand,

catalyze the incorporation of one, or both, oxygen atoms of $O_2$ into their substrate(s), and are consequently referred to as mono- or dioxygenases, respectively. As such, $O_2$ reduction typically co-occurs with substrate oxidation and often requires external electron donors, such as NAD(P)H. Oxygenases can catalyze a broader range of substrates, including aromatic hydrocarbons and fatty acids, and are primarily involved in biosynthesis and detoxification (Bugg 2001; Bernhardt 2006; van Berkel et al. 2006). For example, cytochrome P450 enzymes represent a superfamily of monooxygenases found in all domains of life, which

play a vital role in the biosynthesis of steroids, fatty acids, and bile acids, as well as the inactivation of drugs, toxins, and environmental pollutants (Guengerich 2007). To the best of our knowledge, enzymatic [18]O-KIEs have been experimentally determined for only 26 $O_2$-consuming enzymes, with values ranging from 1.009 to 1.053 (Guy et al. 1989; Cheah et al. 2014, see full list of references in Table S1 of the supplement). This range in enzymatic [18]O-KIEs is equivalent to a range in [18]ε values of -9 ‰ to -50 ‰, significantly exceeding the previously mentioned range of [18]O-ε values observed for respiratory $O_2$

consumption (Mader et al. 2017). Most of these enzymatic [18]O-KIEs have been determined with the primary goal to understand specific enzymatic reaction mechanisms of $O_2$ reduction and substrate oxidation. Comprehensive investigations into the O-isotopic fractionation of enzymatic $O_2$ consumption, which specifically aim at understanding the underlying causes of the observed variability in [18]O-KIEs, are lacking.

To expand and improve our understanding of the variability in isotopic fractionation of $O_2$ at the enzyme level, this

study reports 19 experimentally determined [18]O-KIEs for nine flavin-dependent, five copper-dependent, and one copper-heme-dependent oxidase, as well as for one flavin-dependent monooxygenase. In a first step, enzyme assays were conducted to determine initial $O_2$ consumption rates and Michaelis-Menten kinetic constants of each enzymatic reaction to establish saturating substrate concentrations and the presence or absence of product or substrate inhibition. Subsequently, experiments to determine characteristic [18]O-KIEs were carried out under optimized conditions for each enzyme, whenever possible. For




selected enzymes, additional [18]O-KIEs were measured using alternative substrates, or under limiting $O_2$ concentrations, to assess the influence of these variables on the variability of single-enzyme [18]O-KIEs. The combined analysis of [18]O-KIEs of $O_2$-consuming enzymes determined in this and previous studies allowed a comprehensive assessment of the variability of isotope effects both within the same active-site structure and across different active-site structures. Our findings not only improve the interpretation and generalization of isotopic fractionation of $O_2$ at the enzyme level, but also contribute to a deeper

understanding of the origins of variations in $O_2$ isotopic fractionation at the organism and environmental levels. Ultimately, this research supports the application of stable $O_2$-isotope analysis as a useful and robust tool for investigating $O_2$-biogeochemical dynamics from molecular to ecosystem scales.

## 2 Materials and Methods

### 2.1 Chemicals and enzymes

Unless noted otherwise, enzymes (see list in Table 1) and chemicals were purchased from Sigma-Aldrich and used as received. Sodium phosphate dibasic ($Na_2HPO_4$, 99%, Carl Roth), sodium phosphate monobasic dihydrate ($NaH_2PO_4 \cdot 2H_2O$, 99%, Merck), sodium acetate (98.5%, Carl Roth), 2-amino-2-(hydroxymethyl)-1,3-propanediol (Tris, 99%), *N*-2-hydroxyethylpiperazine-*N'*-2-ethane-1-sulphonic acid sodium salt (HEPES, 99%, Carl Roth), sodium hydroxide (NaOH, 98%), and hydrochloric acid (HCl, 37%, VWR) were used to make buffer solutions. Sodium chloride (NaCl, 99.5%, Carl Roth),

potassium chloride (KCl, 99%), thiamine diphosphate (95%), manganese sulfate ($MnSO_4$, 99%, Carl Roth), flavin adenine dinucleotide disodium salt hydrate (FAD, 95%), DL-dithiothreitol (99%), Thesit® (non-ionic surfactant for membrane research), and isopropanol (HPLC grade, Carl Roth) were added to certain enzyme assays to increase enzymatic activity or substrate solubility. Methanol (99.9%, Carl Roth), ethanol (99.8%, Honeywell), L-ascorbic acid (98%), bilirubin (98%), cholesterol (99%), choline chloride (98%), cytochrome-*c* from bovine heart (95%), D-alanine (98%), histamine

dihydrochloride (99%), D-(+)-glucose (99.5%), D-(+)-mannose (99%), L-kynurenine (98%), *β*-nicotinamide adenine dinucleotide phosphate reduced tetrasodium salt hydrate (NADPH, 95%), hydroquinone (99%), 2,2′-Azino-bis(3-ethylbenzothiazoline-6-sulfonic acid) diammonium salt (ABTS, 98%), L-(+)-lactic acid (98%), L-lysine monohydrochloride (99.5%), sodium pyruvate (99%), and sarcosine (98%) were used as (co)substrates. Hydrogen peroxide ($H_2O_2$, 30%), formaldehyde (36%), acetaldehyde (99.5%), betaine (98%), ammonium chloride (≥99%), *p*-benzoquinone (98%), sodium

bicarbonate (99.5 %), and glycine (98.5%) were used to test product inhibition of enzymatic activities. Sodium sulfite ($Na_2SO_3$, 98%) was used to calibrate optical oxygen sensors. All solutions were made in ultrapure water (18.2 MΩ cm, ELGA LabWater). $O_2$ (99.995%), $N_2$ (99.999%), and He (99.999%) gas were from Carbagas AG.

 

**Table 1. Names, Enzyme Commission (EC) numbers, biological sources, and activities of all enzymes used in this study.**

| Enzyme name | EC no. | Source | Activity [a] |
|---|---|---|---|
| alcohol oxidase | 1.1.3.13 | *Pichia pastoris* | 24 |
| L-ascorbate oxidase | 1.10.3.3 | *Cucurbita sp.* | 1257 |
| bilirubin oxidase | 1.3.3.5 | *Myrothecium verrucaria* | 33 |
| cholesterol oxidase | 1.1.3.6 | microorganisms | 99 |
| choline oxidase | 1.1.3.17 | *Arthrobacter sp.* | 16-19 [b] |
| cytochrome-c oxidase | 7.1.1.9 | bovine heart | 33 |
| D-amino-acid oxidase | 1.4.3.3 | porcine kidney | 12 |
| diamine oxidase | 1.4.3.22 | porcine kidney | 0.0008-0.0018 [b,c] |
| glucose oxidase | 1.1.3.4 | *Aspergillus niger* | 305 |
| kynurenine 3-monooxygenase | 1.14.13.9 | *Pseudomonas fluorescens* | 7000000 [c] |
| laccase | 1.10.3.2 | *Agaricus bisporus* | 32 [c] |
| laccase | 1.10.3.2 | *Trametes versicolor* | 0.9 [c] |
| L-lactate oxidase | 1.1.3.2 | *Aerococcus viridians* | 40 [c] |
| L-lysine oxidase | 1.4.3.14 | *Trichoderma viride* | 39 |
| pyruvate oxidase | 1.2.3.3 | *Aerococcus sp.* | 89 |
| sarcosine oxidase | 1.5.3.1 | *Bacillus sp.* | 50 [c] |

[a] in µmol min$^{-1}$ (mg protein)$^{-1}$ (unless indicated otherwise) determined under specific conditions defined by the manufacturer
[b] multiple batches of enzyme with different activities were used
[c] activity is reported per mg total solid instead of per mg protein

## 2.2 Enzyme assays for kinetic parameters

To measure (initial) $O_2$ consumption rates, enzyme assays were performed in clear glass, crimp-top vials with a volume of 9 mL when closed. These vials contained small magnetic stir bars, were filled headspace-free with assay solution, and closed with hollow butyl rubber stoppers and crimp caps. Assay solutions consisted of an air-equilibrated buffer, an organic substrate, cofactors and co-substrates if necessary, and the respective enzyme of interest (see Appendix A for details). Once filled and closed, vials were placed on a magnetic stirring plate at room temperature ($23 \pm 1$ °C). Enzymatic reactions were initiated with the addition of small volumes of enzyme or substrate solution through the septum with a gas-tight glass syringe. Dissolved $O_2$ concentrations were continuously monitored inside the closed vials with fiber-optic oxygen minisensors and a FireSting meter (PyroScience GmbH) with automated pressure, humidity, and temperature correction. The fiber-optic minisensors are housed in stainless-steel needles (1.1 mm o.d.), with which the crimp vial septa can be pierced. Optical oxygen sensors were calibrated for maximum and minimum dissolved $O_2$ concentrations with air-equilibrated water and with a 300 mM $Na_2SO_3$ solution, respectively. Accurate temperature compensation was performed with optical temperature sensor spots (PyroScience GmbH) inside the vials. These sensor spots were regularly calibrated with the temperature probe of the FireSting meter.



With this type of enzyme assay, initial $O_2$ consumption rates were measured to determine $K_m$ values for all enzymes with varying initial organic substrate concentrations, referred to as $K_m(S)$, except for cytochrome-$c$ oxidase and kynurenine 3-monooxygenase (KMO) because of limited substrate availability. In addition, this type of enzyme assay was used to measure initial $O_2$ consumption rates in presence or absence of reaction products (see Appendix A for details). Inhibition of enzymatic activities due to the presence of reaction products (i.e., product inhibition) was tested for all enzymes, but only detected for KMO and laccase from *Trametes versicolor*, with 2,2'-azino-bis(3-ethylbenzothiazoline-6-sulfonic acid) diammonium salt (ABTS) as the substrate, at relevant product concentrations. Due to this observed product inhibition, $K_m$ values with varying initial $O_2$ concentrations, referred to as $K_m(O_2)$, were determined as described for $K_m(S)$ above for KMO and laccase from *T. versicolor* with ABTS as the substrate (see Appendix A for details). Varying initial $O_2$ concentrations were achieved by mixing air-equilibrated buffer ($270 \pm 10$ µM $O_2$) with $N_2$-purged buffer (approx. 0 µM $O_2$) or $O_2$-purged buffer ($1200 \pm 100$ µM $O_2$). For all other enzymes, $K_m(O_2)$ values were determined from complete $O_2$ consumption experiments performed with the same type of enzyme assay either in air-equilibrated or $O_2$-purged buffer.

## 2.3 Enzyme assays for $^{18}O$-KIEs and λ values

Enzyme assays to determine $^{18}O$-KIEs and λ values were performed in air-equilibrated buffer solutions with saturating concentrations of all other substrates (see Appendix A for details). As saturating substrate concentrations, we considered either 10 times the $K_m(S)$ value or a sufficiently high substrate concentration to limit the difference between the initial and final reaction rate ($v$, determined with Eq. (8) and the corresponding $K_m(S)$ value) in the experiment to below 5 %. These enzyme assays were typically conducted in a 50 mL gas-tight glass syringe equipped with an optical oxygen sensor spot (PyroScience GmbH), an optical temperature sensor spot, and a small magnetic stir bar. Optical sensor spots were placed on the inside wall of the syringe, as close to the Luer-Lock tip as possible, and calibrated as described above for the fiber-optic oxygen sensors. These sensor spots allowed for a continuous, temperature-corrected measurement of $O_2$ concentrations through the glass wall via an optical fiber. The syringe was filled completely with a buffer solution containing all required substrates. To start the reaction, a small volume of enzyme solution was added through the Luer-Lock tip with a gas-tight glass syringe. Immediately after enzyme addition, a stainless-steel needle (0.8 mm o.d.) was attached to the Luer-Lock tip. To limit exchange of $O_2$ with the atmosphere, the needle was flushed with a few drops of assay solution and then pushed into a 12 mm thick chlorobutyl stopper. For experiments with diamine oxidase, the reaction was initiated by adding a small volume of substrate solution to assay solutions already containing the enzyme. Except during sampling, the syringe was placed on a magnetic stirring plate. Six sampling time-points ($t_1$-$t_6$) were determined from the continuously monitored $O_2$ concentrations, typically at 200, 150, 120, 90, 70, and 50 µM remaining $O_2$, corresponding to approx. 25-80 % $O_2$ consumption. To sample, the needle was removed from the stopper and the first mL assay solution was discarded. The next 3-7 mL (depending on $O_2$ concentration) were injected into 12 mL Exetainers (Labco Limited). Before starting an enzyme assay, Exetainers were closed with chlorobutyl septa, purged with He gas for 1 hour, and amended with 100-200 µL of 2 M NaOH or 2-3 M HCl, to stop enzymatic reactions in the added sample. To ensure equal headspace pressure in the Exetainers despite different sample volumes, Exetainer septa were



pierced with a stainless-steel needle (0.45 mm o.d.) connected with a T-piece to a slow He flow, and an open outlet submerged under 10 cm of water during sample injection. After sample injection, Exetainers were shaken and stored upside down until isotope analysis (see section 2.3). Procedural blanks were prepared by transferring 1-7 mL $N_2$ -purged water with a 50 mL gas-tight glass syringe from closed, over-pressured serum bottles into He -purged Exetainers containing NaOH or HCl solution, as

described above for enzyme assay samples. Similarly, quantification standards (see section 2.3 for details) were prepared by transferring 1-5 mL air-equilibrated water with a 50 mL gas-tight glass syringe into He-purged Exetainers. For each experiment, one or more control samples were prepared by transferring 3 mL leftover assay solution without enzyme with a 10 mL gas-tight glass syringe into a He-purged Exetainer containing NaOH or HCl solution. These control samples were used to determine the concentration and isotopic composition of $O_2$ at the start of the experiments ($t_0$).

Some enzyme assays with choline, diamine, and glucose oxidase were also performed in 4-10 identically prepared 12 mL crimp-top vials per assay, as described recently (de Carvalho et al., (2024). Reactions were initiated by injecting a small volume of enzyme or substrate solution through the septa into filled vials. Prior to sampling, a fiber-optic oxygen microsensor (PyroScience GmbH) housed in a stainless-steel needle (0.5 mm o.d.) was inserted through the septa into the vials to measure the remaining $O_2$ concentration. The oxygen sensor was calibrated as described above. After initiating the reaction and before

measuring $O_2$ concentrations, vials were shaken vigorously. To stop reactions at the desired degrees of $O_2$ consumption, 3-7 mL assay solution was transferred into He-purged Exetainers that had been amended with 100-200 µL 2 M NaOH or 2-3 M HCl. Procedural blanks, control samples, and quantification standards were prepared as described above. Experiments with diamine, choline and glucose oxidase performed with the two different setups resulted in equal $^{18}O$-KIEs and λ values, respectively, within error.

All samples, blanks, quantification standards, and controls were placed upside down on an orbital shaker at 125 rpm for 1 h, prior to analysis by gas chromatography coupled to isotope ratio mass spectrometry (GC-IRMS).

**2.4 Stable isotope analysis of $O_2$**

$δ^{18}O$ and $δ^{17}O$ values of $O_2$ were measured in the headspace of Exetainers with a GasBench II coupled via a Conflo IV to a Delta V Plus isotope-ratio mass spectrometer (Thermo Fisher Scientific) as described recently (de Carvalho et al., 2024) and

reported as permil (‰ ± one standard deviation) deviation relative to the international measurement standard Vienna Standard Mean Ocean Water (VSMOW) according to Eq. (5),

$$δ^hO = \left( \frac{(^hO/^lO)_{sample}}{(^hO/^lO)_{VSMOW}} - 1 \right) \tag{5}$$

where $(^hO/^lO)_{sample}$ is the ratio of heavy ($^{18}O$ or $^{17}O$) to light ($^{16}O$) isotopes in $O_2$ in a sample and $(^hO/^lO)_{VSMOW}$ is the ratio of heavy to light O isotopes in VSMOW. Briefly, seven 100 µL injections were made from each Exetainer headspace onto a 60

m Rt-Molsieve 5 Å PLOT column (Restek from BGB Analytik, 0.32 mm ID, 30 µm film thickness) kept at 25°C. Each



GC/IRMS sequence consisted of 5-14 samples from enzyme assays, 10-12 procedural blanks, 5 quantification standards, and 3 air standards. Half of the blanks were measured at the beginning of the sequence, the other half at the end. Air standards were evenly distributed across the sequence and consisted of 150 μL ambient air in 12 mL He. Air standards were used to verify instrument drift (which was never observed), and to perform a one-point calibration of the δ values to the VSMOW

scale. The $\delta^{18}O$ and $\delta^{17}O$ values of $O_2$ in air were assumed to be 23.8 ‰ and 12.1 ‰, respectively (Luz and Barkan 2011; Laskar et al. 2019; Wostbrock et al. 2020). We recently showed, that for $\delta^{18}O$ values, a one-point calibration is sufficient, while for $\delta^{17}O$ values an additional correction factor must be used (de Carvalho et al., 2024). Procedural blanks were used to correct the measured δ values for blank contributions (Pati et al., 2016). Quantification standards were used to relate IRMS peak amplitudes to dissolved $O_2$ concentrations, and to correct δ values for instrument linearity (change in δ values with signal

size) (Werner and Brand 2001).

## 2.5 Data analysis

Initial $O_2$ consumption rates were determined through linear regressions of the continuously measured $O_2$ concentrations versus time during the initial, linear phase of enzyme assays. $^{18}O$-KIEs and λ values were obtained from a single linear regression of all $O_2$ isotope and concentration data from duplicate or triplicate enzyme assays according to Eqs. (6) and (7), respectively.

$$\ln\left(\frac{\delta^{18}O + 1}{\delta^{18}O_0 + 1}\right) = \left(\frac{1}{^{18}O\text{-KIE}} - 1\right) \cdot \ln\left(\frac{[O_2]}{[O_2]_0}\right) \tag{6}$$

$$\ln(\delta^{17}O + 1) = \lambda \cdot \ln(\delta^{18}O + 1) \tag{7}$$

where $[O_2]_0$ and $\delta^{18}O_0$ are the initial concentration and $\delta^{18}O$ value of $O_2$, respectively, measured in the control sample (see
section 2.2), and $[O_2]$, $\delta^{18}O$, and $\delta^{17}O$ are the values measured in each enzyme assay sample at the different time points. All linear regressions were performed with Microsoft Excel, and errors are reported as 95 % confidence intervals. $K_m$ values were determined with a non-linear least square regression according to Eq. (8),

$$\upsilon_t = \frac{\upsilon_{max} \cdot [i]_t}{K_m(i) + [i]_t} \tag{8}$$

where $\upsilon_t$ is the $O_2$ consumption rate at a given time point t, $\upsilon_{max}$ is the maximum $O_2$ consumption rate of an enzymatic reaction,
$[i]_t$ is the concentration of an organic substrate (S) or $O_2$ at time t, and $K_m(i)$ is the Michaelis constant determined under constant initial $O_2$ and variable initial substrate concentration ($K_m(S)$), or under constant initial substrate and variable initial $O_2$ concentration ($K_m(O_2)$). For all $K_m(S)$, as well as for $K_m(O_2)$ values determined for KMO and laccase from *T. versicolor* with ABTS, regressions were performed with initial rates of $O_2$ consumption ($\upsilon_0$) from different experiments against the nominal initial organic substrate concentrations ($[S]_0$), or against the measured initial $O_2$ concentrations ($[O_2]_0$), respectively. For all





other enzymes, where product inhibition was not detected, we determined $K_m(O_2)$ values from the continuous measurement of $O_2$ concentration over time ($[O_2]_t$) in a single enzyme assay, as described previously (Pati et al. 2022). For each time-point, $v_t$ was calculated as the derivative of the measured $[O_2]_t$ vs. t (i.e., $\Delta[O_2]_t/\Delta t$) with Igor Pro software (WaveMetrics, Inc.). $K_m$ values and corresponding 95 % confidence intervals were determined with R software (R Core Team 2023) using the MASS package (Venables and Ripley 2002).

**3 Results**

**3.1 Kynurenine 3-monooxygenase**

The flavin-dependent KMO was studied as an example for flavin monooxygenases, for which $O_2$ reduction mechanisms have been well-described. $K_m(S)$ values for the native substrate L-kynurenine ($0.012 \pm 0.003$ mM) and the co-substrate NADPH ($0.009 \pm 0.001$ mM) were obtained from literature (Crozier and Moran 2007). $K_m(O_2)$ and $^{18}$O-KIE were determined in

experiments at optimal pH (7.5) and room temperature ($23 \pm 1$ °C), with saturating concentrations (see section 2.2 for details) of L-kynurenine (1 mM) and NADPH (0.5 mM), as well as 2 mM dithiothreitol to prevent loss of KMO activity (Crozier and Moran 2007). A $K_m(O_2)$ of $6 \pm 4$ µM was determined from initial rates of $O_2$ consumption measured in 10 separate experiments with different initial $O_2$ concentrations (25-260 µM) as shown in Fig. 1A. The $^{18}$O-KIE and $\lambda$ values were determined from the change in concentration, $\delta^{18}$O, and $\delta^{17}$O of $O_2$ over time, measured in duplicate experiments. Figs. 1B and 1C illustrate typical

$\delta^{18}$O data from one experiment. The combined data from both experiments (see section 2.4 for details) resulted in a $^{18}$O-KIE of $1.0304 \pm 0.0003$ and a $\lambda$ value of $0.545 \pm 0.005$.

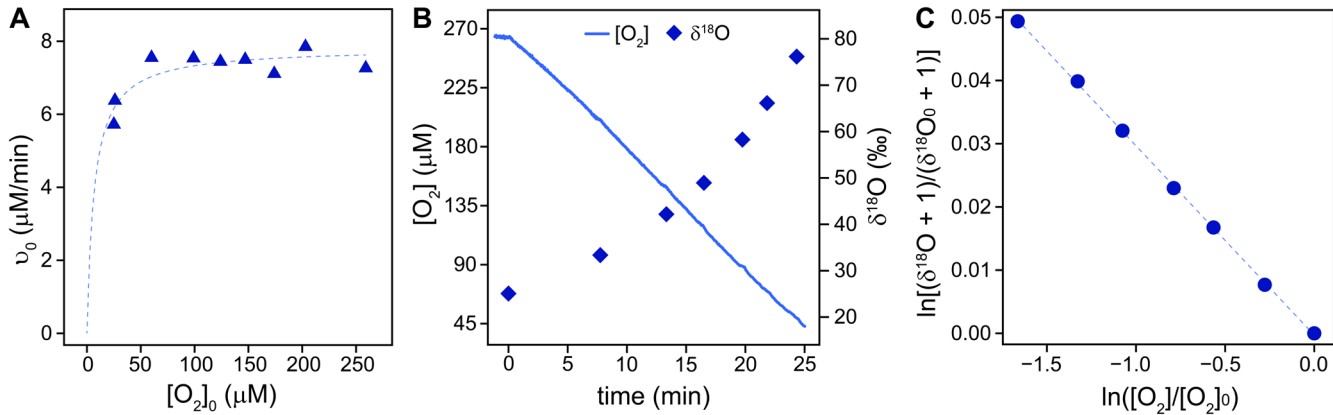

**Figure 1. A) Initial rates of $O_2$ consumption ($v_0$) by KMO (blue triangles) measured in 10 separate experiments with different initial $O_2$ concentrations ($[O_2]_0$). The dotted line illustrates a non-linear least square regression fit according to Eq. (8), which was used to**

**obtain $K_m(O_2)$. B) Continuously measured $O_2$ concentrations (solid blue line) and $\delta^{18}$O values of $O_2$ measured in discrete samples (blue diamonds) over time during an experiment with KMO. C) Linearized and normalized data ($\delta^{18}$O vs. $[O_2]$) from Fig. 1B, where $[O_2]_0$ and $\delta^{18}O_0$ represent the concentration and $\delta^{18}$O value of $O_2$ at the beginning of the experiment. The dotted line shows a linear regression fit according to Eq. (6), from which the $^{18}$O-KIE was obtained.**





### 3.2 Flavin-dependent oxidases

Nine flavin-dependent oxidases were investigated. All of them convert $O_2$ to $H_2O_2$ in the oxidative half-reaction and oxidize an organic substrate in the reductive half-reaction. Pyruvate oxidase was the only flavin-dependent oxidase that required cofactors for activity, namely thiamine diphosphate, $MnSO_4$, and FAD. Experiments with cholesterol oxidase were performed with the surfactant Thesit® and isopropanol due to the low water solubility of the native substrate cholesterol. Experiments with glucose and alcohol oxidase were each performed with their native and an alternative substrate: Glucose and mannose, in

the case of glucose oxidase, and methanol and ethanol, the case of alcohol oxidase. Experiments to determine $^{18}O$-KIEs for alcohol, choline, and L-lysine oxidase were performed at two different initial $O_2$ concentrations ($260 \pm 10$ µM and $1200 \pm 100$ µM).

### 3.2.1 Michaelis constants for organic substrates

    $K_m(S)$ values were determined at $260 \pm 10$ µM initial $O_2$ concentration, as described for the $K_m(O_2)$ value of KMO (see section

3.1). However, initial rates of $O_2$ consumption were measured at different initial organic substrate concentrations. For all flavin-dependent oxidases, except pyruvate oxidase, $K_m(S)$ values were determined for the native substrate, with values ranging from $0.011 \pm 0.004$ mM for L-lysine oxidase to $36 \pm 18$ mM for glucose oxidase (see Table 2). The $K_m(S)$ for pyruvate oxidase could not be determined as the initial rates of $O_2$ consumption were not linear across all relevant pyruvate concentrations. For alcohol oxidase, the alternative substrate ethanol had a substantially higher $K_m(S)$ than the native substrate methanol ($22 \pm 6$

vs. $0.6 \pm 0.4$ mM). For D-mannose, the alternative substrate of glucose oxidase, a $K_m(S)$ could not be determined because initial rates of $O_2$ consumption increased linearly with D-mannose concentrations up to the solubility limit of D-mannose.

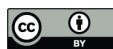



**Table 2.** $K_m(S)$, $K_m(O_2)$, $^{18}O$-KIEs and λ values determined for all enzymes investigated in this study with errors given as 95 % confidence intervals.

| Active site | Enzyme | Substrate | $K_m(S)$ (mM) | | $K_m(O_2)$ (µM) | | $^{18}O$-KIE (-) | | λ (-) | |
|---|---|---|---|---|---|---|---|---|---|---|
| Flavin | kynurenine 3-monooxygenase | L-kynurenine | n.d.[a] | | 6 ± 4 | | 1.0304 ± 0.0003 | | 0.545 ± 0.005 | |
| Flavin | alcohol oxidase | methanol | 0.6 ± 0.4 | | 1017 ± 93 | | 1.028 ± 0.001 | | 0.491 ± 0.008 | |
| Flavin | alcohol oxidase | ethanol | 22 ± 6 | | 901 ± 200 | | 1.0276 ± 0.0007 | | 0.483 ± 0.007 | |
| Flavin | cholesterol oxidase | cholesterol | 0.3 ± 0.2 | | 271 ± 12 | | 1.0191 ± 0.0003 | | 0.53 ± 0.01 | |
| Flavin | choline oxidase | choline | 0.5 ± 0.1 | | 312 ± 21 | | 1.0194 ± 0.0006 | | 0.537 ± 0.008 | |
| Flavin | D-amino acid oxidase | D-alanine | 2.3 ± 0.4 | | 92 ± 7 | | 1.0509 ± 0.0008 | | 0.546 ± 0.004 | |
| Flavin | glucose oxidase | D-glucose | 36 ± 18 | | 116 ± 14 | | 1.029 ± 0.001 | | 0.523 ± 0.009 | |
| Flavin | glucose oxidase | D-mannose | n.d.[a] | | 3.9 ± 0.5 | | 1.0341 ± 0.0005 | | 0.536 ± 0.004 | |
| Flavin | L-lactate oxidase | L-lactate | 0.3 ± 0.1 | | 80 ± 3 | | 1.044 ± 0.001 | | 0.540 ± 0.006 | |
| Flavin | L-lysine oxidase | L-lysine | 0.011 ± 0.004 | | 1291 ± 73 | | 1.046 ± 0.001 | | 0.543 ± 0.004 | |
| Flavin | pyruvate oxidase | pyruvate | n.d.[a] | | 225 ± 16[b] | | 1.0565 ± 0.0009 | | 0.547 ± 0.002 | |
| Flavin | sarcosine oxidase | sarcosine | 8 ± 3 | | 83 ± 3 | | 1.047 ± 0.001 | | 0.536 ± 0.007 | |
| Copper | L-ascorbate oxidase | L-ascorbic acid | 0.14 ± 0.05 | | 144 ± 11 | | 1.0086 ± 0.0006 | | 0.54 ± 0.01 | |
| Copper | bilirubin oxidase | bilirubin | 0.018 ± 0.009 | | 73 ± 3 | | 1.0222 ± 0.0005 | | 0.535 ± 0.009 | |
| Copper | diamine oxidase | histamine | 0.018 ± 0.007 | | 9.4 ± 0.5 | | 1.0103 ± 0.0007 | | 0.51 ± 0.03 | |
| Copper | laccase from *A. bisporus* | hydroquinone | 2 ± 3 | | 36 ± 2 | | 1.0190 ± 0.0002 | | 0.530 ± 0.007 | |
| Copper | laccase from *T. versicolor* | hydroquinone | 0.23 ± 0.04 | | 72 ± 4 | | 1.0196 ± 0.0007 | | 0.539 ± 0.007 | |
| Copper | laccase from *T. versicolor* | ABTS | 0.12 ± 0.07 | | 47 ± 59 | | 1.0194 ± 0.0005 | | 0.54 ± 0.01 | |
| copper/ heme | cytochrome-*c* oxidase | cytochrome *c* | n.d.[a] | | 3.3 ± 0.5 | | 1.0189 ± 0.0005 | | 0.543 ± 0.009 | |

[a] not determined
[b] tentative value (see section 3.2.2)





### 3.2.2 Michaelis constants for $O_2$

$K_m(O_2)$ values were determined from complete $O_2$ consumption experiments at saturating organic substrate concentrations (see section 2.2), as shown in Figs. 2A and 2B for L-lactate oxidase as an example. Three flavin-dependent oxidases exhibited $K_m(O_2)$ values exceeding air-saturated $O_2$ concentrations in the presence of their native substrates, namely alcohol oxidase with both substrates ($1017 \pm 93$ μM and $901 \pm 200$ μM), choline oxidase ($312 \pm 21$ μM), and L-lysine oxidase ($1291 \pm 73$ μM). For these enzymes, $K_m(O_2)$ values were obtained from complete $O_2$ consumption experiments with initial $O_2$ concentrations of $1200 \pm 100$ μM. The remaining flavin-dependent oxidases displayed $K_m(O_2)$ values between $80 \pm 3$ μM and $260 \pm 12$ μM (see Table 2). The $K_m(O_2)$ value determined for pyruvate oxidase ($225 \pm 16$ μM) should be considered a tentative value as the effect of product inhibition could not be assessed, and the $K_m(S)$ could not be determined. $K_m(O_2)$ values for alcohol and glucose oxidase were also determined at saturating concentrations of the alternative substrates, ethanol and D-mannose, respectively. In the case of alcohol oxidase, the $K_m(O_2)$ values determined with methanol ($1017 \pm 93$ μM) and ethanol ($901 \pm 200$ μM) as substrates were equal within error. In contrast, glucose oxidase exhibited a significantly lower $K_m(O_2)$ value with D-mannose as the substrate ($3.9 \pm 0.5$ μM) compared to the value determined with the native substrate D-glucose ($116 \pm 14$ μM).

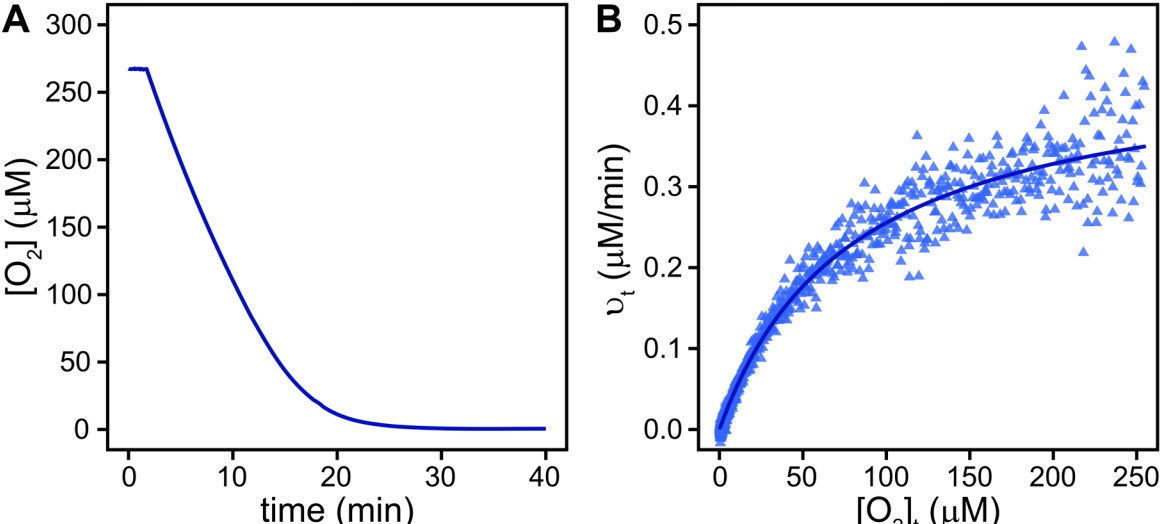

**Figure 2. A) $O_2$ concentration ($[O_2]$) over time in a complete $O_2$ consumption experiment with L-lactate oxidase. B) Blue triangles show reactions rates ($v_t$), derived by differentiating the data in Fig. 2A at corresponding $O_2$ concentrations ($[O_2]_t$). The solid line shows a non-linear least square regression fit according to Eq. (8).**





### 3.2.3 $^{18}$O-Kinetic isotope effects and λ values

All $^{18}$O-KIEs were determined in air-saturated buffer solutions with saturating native substrate concentrations, as described for KMO (see section 3.1). The $^{18}$O-KIEs of D-amino-acid, L-lactate, L-lysine, pyruvate, and sarcosine oxidase ranged from

$1.044 \pm 0.001$ to $1.0565 \pm 0.0009$ (see Table 2). In contrast, alcohol, cholesterol, choline, and glucose oxidase were associated with lower $^{18}$O-KIEs, ranging from $1.0191 \pm 0.0003$ to $1.029 \pm 0.001$ (see Table 2). Because alcohol, choline, and L-lysine oxidase exhibited $K_m(O_2)$ values above air-saturation, their $^{18}$O-KIEs were additionally determined in $O_2$-purged buffer with initial $O_2$ concentrations of $1200 \pm 100$ µM (see Appendix B for details). For all three enzymes, the $^{18}$O-KIEs were identical, within error, irrespective of the initial $O_2$ concentration (data not shown). The $^{18}$O-KIEs of alcohol oxidase with the two

substrates methanol and ethanol were also identical within error (see Table 2). However, the $^{18}$O-KIE determined for glucose oxidase with D-mannose was larger ($1.0341 \pm 0.0005$) than that determined with D-glucose ($1.029 \pm 0.001$). λ values ranged between $0.523 \pm 0.009$ and $0.547 \pm 0.002$ (see Table 2) for all flavin oxidases except for alcohol oxidase, which yielded lower λ values of $0.491 \pm 0.008$ and $0.483 \pm 0.007$ with methanol and ethanol, respectively.

### 3.3 Copper-dependent oxidases

Five copper-dependent oxidases were investigated. Namely, Laccases, L-ascorbate oxidase, and bilirubin oxidase, which convert $O_2$ to water in the oxidative half-reaction, and diamine oxidase, which converts $O_2$ to $H_2O_2$. All experiments were performed with native substrates, except experiments with laccase (see below), in a buffered, air-equilibrated solution at optimal pH and room temperature.

     Laccases are multicopper oxidases that can oxidize a wide variety of substrates and lack a specific native substrate

(Strong and Claus 2011). Hydroquinone and ABTS were selected as substrates in this study because they displayed different substrate-to-$O_2$ consumption stoichiometries. Four ABTS molecules are required to reduce one molecule of $O_2$, while only two hydroquinone molecules are required to reduce one molecule of $O_2$ (see Appendix C). Despite these differences, laccase from *T. versicolor* yielded similar values for $K_m(S)$, $K_m(O_2)$, and $^{18}$O-KIEs irrespective of the substrate oxidized. With hydroquinone as the substrate, $K_m(S)$, $K_m(O_2)$, and $^{18}$O-KIE were $0.23 \pm 0.04$ mM, $72 \pm 4$ µM, and $1.0196 \pm 0.0007$,

respectively. With ABTS as the substrate, $K_m(S)$, $K_m(O_2)$, and $^{18}$O-KIE were $0.12 \pm 0.07$ mM, $47 \pm 59$ µM, and $1.0194 \pm 0.0005$, respectively. Laccase from *Agaricus bisporus* exhibited a 10-fold higher $K_m(S)$ and a 2-fold lower $K_m(O_2)$ with hydroquinone as the substrate compared to laccase from *T. versicolor* under identical conditions (see Table 2). However, the $^{18}$O-KIEs were identical within error ($1.0190 \pm 0.0002$) and λ values ranged from $0.530 \pm 0.007$ to $0.54 \pm 0.01$ (see Table 2).

     The remaining three copper-dependent oxidases, L-ascorbate, bilirubin, and diamine oxidase displayed low $K_m(S)$

values between $0.14 \pm 0.05$ mM and $0.018 \pm 0.007$ mM (see Table 2). $K_m(O_2)$ values decreased from $144 \pm 11$ µM for L-ascorbate oxidase to $73 \pm 3$ µM for bilirubin oxidase and $9.4 \pm 0.5$ µM for diamine oxidase. L-Ascorbate and diamine oxidase exhibited the lowest observed $^{18}$O-KIEs of all enzymes in this study with $1.0086 \pm 1.0006$ and $1.0103 \pm 0.0007$, respectively, while bilirubin oxidase had an $^{18}$O-KIE of $1.0223 \pm 1.0005$. λ values ranged from $0.51 \pm 0.03$ to $0.54 \pm 0.01$ (see Table 2).



During experimental assays with diamine oxidase, $O_2$ production due to catalase contamination in the lyophilized diamine oxidase powder was detected. Catalase catalyzes the oxidation of $H_2O_2$ to $H_2O$ and $O_2$, which could lead to inaccurate measurements of $O_2$ consumption by diamine oxidase. To address this potential interference, $^{18}O$-KIEs for diamine oxidase were determined in the presence of the catalase contamination alone and with the addition of excess horseradish peroxidase and ascorbic acid. Horseradish peroxidase catalyzes the oxidation of $H_2O_2$ and ascorbic acid to $H_2O$ and dehydroascorbic acid. In the presence of excess horseradish peroxidase, $H_2O_2$ was converted to $H_2O$ faster than catalase could reduce $H_2O_2$ to $H_2O$ and $O_2$. The $^{18}O$-KIEs determined for diamine oxidase were found to be identical within error, regardless of catalase activity (data not shown).

### 3.4 Cytochrome-*c* oxidase

Cytochrome-*c* oxidase is a heme-copper dependent-oxidase, in which the heme $a_3$ subunit initially binds $O_2$ (Yoshikawa and Shimada 2015). The $K_m(S)$ was not determined, but all experiments were performed with 25 μM cytochrome *c* and 3 mM ascorbic acid, to continuously reduce the product ferricytochrome *c* back to the substrate ferrocytochrome *c*. $K_m(S)$ values for ferrocytochrome *c* are reported to be 1.48 μM or lower (Merle and Kadenbach 1982). Therefore, the 25 μM of cytochrome *c* used is considered a saturating substrate concentration. The $K_m(O_2)$ was $3.3 \pm 0.5$ μM, the $^{18}O$-KIE was $1.0189 \pm 0.0005$, and the λ value was $0.543 \pm 0.009$ (see Table 2).

### 4 Discussion

#### 4.1 $^{18}O$-KIEs of flavin-dependent $O_2$-consuming enzymes

Flavin-dependent $O_2$-consuming enzymes utilize derivatives of the vitamin riboflavin as cofactors in their active sites. The organic flavin cofactor can be present in three different redox states: fully oxidized flavin (FL), radical flavin intermediate (FLH$^{\bullet}$), and fully reduced flavin (FLH$_2$ or FLH$^-$). The oxidation of FLH$_2$ to FL releases two electrons and two protons, which can be used for the reduction of $O_2$ (Massey 2002), as illustrated in a simplified catalytic cycle in Fig. 3. The reduction of $O_2$ by both flavin-dependent monooxygenases and oxidases starts with an outer-sphere single-electron transfer from FLH$_2$ (or FLH$^-$) to $O_2$ forming FLH$^{\bullet}$ and $O_2^{\bullet-}$. A recombination of the two radical species then forms a peroxyflavin intermediate (FLOO$^-$), which can be protonated to a hydroperoxyflavin intermediate (FLOOH). In all known flavin-dependent monooxygenases, the (hydro)peroxyflavin can be detected and is responsible for substrate hydroxylation with concomitant O-O bond cleavage to form a hydroxyflavin (FLOH) (Massey 2002) (see blue arrows in Fig. 3). In a subsequent step, FLOH reacts to FL by releasing $H_2O$ (see blue arrows in Fig. 3). In flavin-dependent oxidases, FLOO(H) has not been observed directly, and its formation remains a matter of ongoing debate (Massey 2002). In addition to FL formation similar to the monooxygenation pathway (FLH$^{\bullet}$ and $O_2^{\bullet-}$ recombination to FLOO(H) and subsequent release of hydrogen peroxide), FL can also be formed through a sequence of outer-sphere electron and proton transfer steps from FLH$^{\bullet}$ to $O_2^{\bullet-}$ without covalent-bond formation between the flavin and $O_2$ (see green arrows in Fig. 3) (Massey 2002; Mattevi 2006; Chaiyen et al. 2012). The reduction of FL





**Figure 3. $O_2$ reduction mechanism of flavin-dependent oxidases and monooxygenases. Black arrows indicate common reaction steps, blue arrows indicate reaction steps performed by monooxygenases, and green arrows indicate reaction steps performed by oxidases. S represents the organic substrate.**

to $FLH_2$ is coupled with substrate or co-substrate oxidation in oxidases and monooxygenases, respectively, to complete the catalytic cycle (see Fig. 3).

In this study, we determined the first $^{18}O$-KIE for a flavin-dependent monooxygenase, namely KMO, which was $1.0305 \pm 0.0003$. The magnitude of this isotope effect indicates that changes in bond order of $O_2$ occur during the rate-limiting step of the reaction between KMO and $O_2$, excluding $O_2$ binding and product release as possible rate-limiting steps. Hence, the rate-limiting reaction step of KMO is the formation of $O_2^{\cdot-}$, $FLOO^-$, FLOOH, or S-OH and FLOH (see blue arrows in Fig. 3). $^{18}O$-EIEs have been calculated for the reversible formation of $O_2^{\cdot-}$, $HO_2^-$, $H_2O_2$, and two $H_2O$ from $O_2$ as 1.033, 1.034, 1.009, and 0.968, respectively (Roth and Klinman 2005). When comparing experimental $^{18}O$-KIEs to calculated $^{18}O$-EIEs, it is generally assumed that a measured $^{18}O$-KIE (i) reflects intrinsic $^{18}O$-KIEs of all electron and proton transfer steps up to, and including, the rate-limiting step and (ii) is similar to, but not larger than, the $^{18}O$-EIE calculated for the formation of the product/intermediate after the rate-limiting step (Roth and Klinman 2005). Based on these $^{18}O$-EIEs, the reduction of $O_2$ by KMO is thus likely characterized by a rate-limiting $O_2^{\cdot-}$ or $FLOO^-$ formation. However, this conclusion conflicts with studies suggesting that substrate hydroxylation is the rate-limiting step (Özkılıç and Tüzün 2019).

Similar magnitudes of $^{18}O$-KIEs compared to KMO have been determined in this study for glucose oxidase: $1.029 \pm 0.001$ and $1.0341 \pm 0.0005$ with D-glucose and D-mannose as the substrate, respectively. These values agree with previous studies of the same enzyme (Su and Klinman 1998). Based on $^{18}O$-EIEs, solvent isotope effects, and viscosity effects, Roth and Klinman, (2003) suggested the initial outer sphere electron transfer from $FLH^-$ to $O_2$ to be the rate-limiting step of $O_2$





reduction by glucose oxidase. The $^{18}$O-KIEs determined for cholesterol, choline, and alcohol oxidase in this study were similar to, or lower than, those determined for KMO and glucose oxidase (1.0191-1.028, see Table 2). Because these isotope effects were still larger than the calculated $^{18}$O-EIE for $H_2O_2$ or $H_2O$ formation (1.009, and 0.968, respectively), these enzymes likely also have a rate-limiting $O_2^{•-}$ or $FLOO^-$ formation. In contrast, D-amino-acid, L-lactate, L-lysine, pyruvate, and sarcosine oxidase exhibited much larger $^{18}$O-KIEs (1.044-1.0565, see Table 2). These distinctively high $^{18}$O-KIEs clearly suggest a

different rate-limiting step than previously discussed, even though the first outer-sphere electron transfer to $O_2$ has also been proposed as the rate-limiting step for D-amino-acid oxidase (Kiss and Ferenczy 2019). To date, $^{18}$O-KIEs of this magnitude have only been measured for L-amino acid (1.0478) and D-amino-acid oxidase (1.053) by Cheah et al. (2014). The only $^{18}$O-EIE of similar magnitude was calculated for the formation of $O_2^{2-}$, a two-electron reduction product of $O_2$, with a value of 1.050 (Roth and Klinman 2003).

Considering the two possible reaction mechanisms described in Fig. 3 for flavin-dependent oxidases, we suggest that glucose, cholesterol, choline, and alcohol oxidase, like KMO, reduce $O_2$ through the formation of FLOO(H) with a rate-limiting formation of either $O_2^{•-}$ or $FLOO^-$. The same applies to glycolate oxidase with a $^{18}$O-KIE of 1.023 (Guy et al., 1993; Ribas-Carbo et al., 1995; Cheah et al., 2014). However, for D-amino-acid, L-amino-acid, L-lactate, L-lysine, pyruvate, and sarcosine oxidase we suggest the alternative $O_2$ reduction mechanism, where FL is formed directly from $FLH^•$ and $O_2^{•-}$ without the

formation of FLOO(H) (see green arrows in Fig. 3). Still, the exact nature of the rate-limiting step (a second single electron transfer, a proton-coupled electron transfer, or a hydrogen atom transfer) in this alternative $O_2$ reduction mechanism cannot be inferred from the current experimental evidence. It is also possible that the rate-limiting step differs among the six oxidases with $^{18}$O-KIEs between 1.044 and 1.057, or that the first electron transfer to $O_2$ is partially rate-limiting in some of these enzymes, which could explain the lower-than-expected $^{18}$O-KIEs for such a rate-limiting step.

For KMO, cholesterol, choline, and glycolate oxidase, as well as glucose oxidase with 45 different substrates, which we consider to share a common reaction mechanism, we found a tentative correlation between $^{18}$O-KIEs and the corresponding $K_m(O_2)$ values (see Fig. 4). The $K_m(O_2)$ values for glucose oxidase with the substrate 2-deoxy-D-glucose and for glycolate oxidase were reported to be $25 \pm 5$ μM and 210 μM, respectively (Macheroux et al. 1991; Roth and Klinman 2003). Based on the limited number of data points, we do not consider the correlation to be necessarily linear as shown in Fig. 4. However, the

data clearly indicates that enzymes with lower $K_m(O_2)$ values have higher $^{18}$O-KIEs, ranging from choline oxidase with a $K_m(O_2)$ of $298 \pm 20$ μM and a $^{18}$O-KIE of $1.0194 \pm 0.0006$, to glucose oxidase with D-mannose as the substrate with a $K_m(O_2)$ of $3.9 \pm 0.6$ μM and a $^{18}$O-KIE of $1.0341 \pm 0.0005$. Since $^{18}$O-KIEs reflect the ratios of reaction rates of the different $O_2$ isotopologues, a correlation between $^{18}$O-KIE and $K_m(O_2)$ only makes sense when we consider the kinetic properties of the Michaelis constant. $K_m(O_2)$ is the ratio of the two apparent rate constants "$v_{max}$" and "$v_{max}/K_m(O_2)$". "$v_{max}$" represents the

observed rate at high substrate concentration and includes all steps after the formation of an enzyme-substrate complex (Northrop 1998). "$v_{max}/K_m(O_2)$" represents the observed rate at low substrate concentration, encompassing all steps beginning with interaction of enzyme with $O_2$ up to, and including, the first rate-limiting step (Northrop 1998). In $O_2$-consuming enzymes, $O_2$ typically binds to the enzyme after binding of the organic substrate (oxygenases), or in a ping-pong mechanism (oxidases)



(Malmstrom 1982; Romero et al. 2018). Thus, the only step covered by "$v_{max}/K_m(O_2)$" but not by "$v_{max}$" is $O_2$ binding.

Therefore, when $K_m(O_2)$ is very large, "$v_{max}/K_m(O_2)$" is much smaller than "$v_{max}$", and $O_2$ binding must be slower than the catalytic step. However, as $K_m(O_2)$ decreases, "$v_{max}/K_m(O_2)$" becomes closer to "$v_{max}$", and $O_2$ binding contributes less to the overall reaction rate. Consequently, $K_m(O_2)$ can be interpreted as a proxy for the extent to which $O_2$ binding contributes to the overall reaction rate. If $O_2$ binding was the sole rate-limiting step, an apparent $^{18}$O-KIE close to 1 would be expected because no bond changes would occur in $O_2$. However, this is not the case for any $O_2$-consuming enzyme studied so far. On the other

extreme, if $O_2$ binding does not contribute to the overall rate at all, the apparent $^{18}$O-KIE is expected to reflect the intrinsic $^{18}$O-KIE of the rate-limiting step. Accordingly, the intrinsic $^{18}$O-KIE for the rate-limiting step of $O_2^{\bullet-}$ or FLOO$^-$ formation is likely between 1.030 and 1.035, based on both calculated $^{18}$O-EIEs for these reactions (1.033-1.034) (Roth and Klinman 2003), and on the maximum $^{18}$O-KIEs observed for glucose oxidase (1.0341 ± 0.0005) and KMO (1.0304 ± 0.0003). The lower $^{18}$O-KIEs (1.019-1.0.23), particularly for cholesterol, choline, and glycolate oxidase, can thus still arise from a rate-limiting $O_2^{\bullet-}$ or

FLOO$^-$ formation, but with increasing contributions from a slower $O_2$ binding to the overall reaction rate.

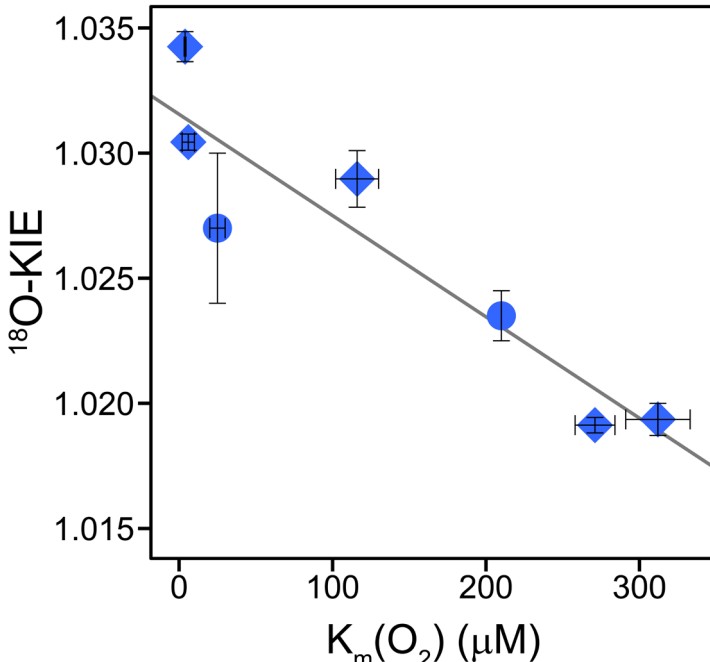

**Figure 4. Correlation of $^{18}$O-KIEs and corresponding Km(O₂) values of glucose, choline, cholesterol, and glycolate oxidase as well as KMO. Blue diamonds represent the $^{18}$O-KIEs and corresponding Km(O₂) values determined in this study. Blue circles represent the $^{18}$O-KIEs and corresponding Km(O₂) values obtained from literature for glycolate oxidase and for glucose oxidase with 2-deoxy-**
**D-glucose as the substrate (Macheroux et al., 1991; Su and Klinman, 1999; Roth and Klinman, 2003; Cheah et al., 2014). Error bars indicate 95 % confidence intervals. The solid line indicates a tentative linear correlation.**





Alcohol oxidase, with either methanol or ethanol as the substrate, was the only enzyme with $^{18}$O-KIEs close to or below 1.03 that did not follow the observed trend between $K_m(O_2)$ values and $^{18}$O-KIEs discussed above. $K_m(O_2)$ values of alcohol oxidase ($1017 \pm 93$ and $901 \pm 200$) were substantially larger than $K_m(O_2)$ values of all other flavin-dependent enzymes studied, except for L-lysine oxidase, which likely has a different $O_2$ reduction mechanism (formation of $O_2^{2-}$). Interestingly, alcohol oxidase was the only enzyme tested in this study that exhibited particularly low λ values between $0.483 \pm 0.007$ and $0.488 \pm 0.009$ (see Table 2). These values are not only lower than typical λ values (0.51-0.53) but also significantly lower than λ values observed for all other enzymes in this study, which ranged from $0.51 \pm 0.03$ to $0.547 \pm 0.002$ (see Table 2). We note that λ values determined for the majority of enzymes in this study are close to, but slightly higher than previously determined λ values of 0.51 to 0.53 for biological $O_2$ consumption (Young et al. 2002; Luz and Barkan 2005; Ash et al. 2020; Hayles and Killingsworth 2022). It is possible that the applied $\delta^{17}O$ scale correction factor from de Carvalho et al., (2024) leads to a slight overestimation of λ values. Regardless of this uncertainty in the $\delta^{17}O$ scale correction factor, the λ values determined for alcohol oxidase are clearly much lower than any previously determined λ values for biological $O_2$ consumption, and significantly lower than those for any other enzyme studied here. This difference in λ values suggests a unique $O_2$ reduction mechanism for alcohol oxidase, differing from the mechanism proposed for enzymes that exhibit a correlation between $^{18}$O-KIEs and $K_m(O_2)$ values. However, this reduction mechanism cannot be further elucidated in this study.

## 4.2 $^{18}$O-KIEs of metal-dependent $O_2$-consuming enzymes

Unlike flavin-dependent $O_2$-consuming enzymes, which have a relatively conserved active site and catalytic mechanism, iron- and copper-dependent $O_2$-consuming enzymes are known to employ a wide variety of different active site structures and catalytic mechanisms (Costas et al. 2004; Blank et al. 2010; Liu et al. 2014; Solomon et al. 2014; Huang and Groves 2018). For the five copper-dependent oxidases tested in this study and for six (out of seven) copper-dependent monooxygenases and oxidases examined in previous research, the $^{18}$O-KIEs grouped closely around two main values. Namely, L-ascorbate and diamine oxidase from this study, as well as tyrosinase, bovine serum amine oxidase, and amine oxidase from *Hansenula polymorpha* were characterized by $^{18}$O-KIEs between 1.0086 and 1.011 (see Tables 1 and S1 in the supplement) (Feldman et al. 1959; Su and Klinman 1998; Welford et al. 2007). Conversely, the $^{18}$O-KIEs of bilirubin oxidase and the two laccases from this study, as well as peptidylglycine monooxygenase, dopamine β-monooxygenase, and galactose oxidase ranged between 1.0173 and 1.223 (see Tables 1 and S1 in the supplement) (Tian et al. 1994; Francisco et al. 2003; Humphreys et al. 2009). The only copper-dependent enzyme studied so far that fell in between these two clusters is pea-seedling amine oxidase with an $^{18}$O-KIE of $1.014 \pm 0.001$ (Mukherjee et al. 2008). The two groups of copper-dependent enzymes defined by the two groups of $^{18}$O-KIE values, both contain a mix of monooxygenases and oxidases (see Fig. 5). The monooxygenases peptidylglycine monooxygenase, dopamine β-monooxygenase, and tyrosinase catalyze the incorporation of one O atom from $O_2$ into their substrate. Multicopper oxidases, including laccase, L-ascorbate oxidase, and bilirubin oxidase, reduce $O_2$ to two $H_2O$. The cofactor-dependent mononuclear copper enzymes (copper amine oxidases including diamine oxidase and galactose oxidase)





reduce $O_2$ to $H_2O_2$ (Mure et al. 2002; Humphreys et al. 2009). Despite these differences, all copper-dependent $O_2$-consuming enzymes form common copper-oxygen intermediates, namely copper-superoxo (Cu(II)-OO•), copper-peroxo (Cu(II)-OO⁻), and copper-hydroperoxo (Cu(II)-OOH) species. Figure 6 shows the electron and proton transfer steps involved in the formation of these intermediates. $^{18}$O-EIEs for the reversible formation of these three copper-oxygen species have been determined to be 1.009-1.010 for copper-superoxo, 1.018-1.031 for copper-peroxo, and 1.025-1.026 for copper-hydroperoxo intermediates

(Mukherjee et al., 2008; Humphreys et al., 2009). The copper-dependent enzymes that exhibited $^{18}$O-KIEs between 1.0173 and 1.223 are thus likely characterized by a rate-limiting step involving the formation of a copper-peroxo or copper-hydroperoxo intermediate. Accordingly, studies of peptidylglycine and dopamine β-monooxygenase, which exhibited $^{18}$O-KIEs of 1.0173 ± 0.0009 and 1.0197 ± 0.0003, respectively, suggested a rate-limiting hydrogen atom abstraction by a copper-superoxo intermediate to form a copper-hydroperoxo species (Evans et al. 2003; Osborne and Klinman 2011). The $^{18}$O-KIE

of 1.019 ± 0.001 determined for galactose oxidase by Humphreys et al., (2009) was also attributed to a rate-limiting hydrogen atom abstraction by a copper-superoxo intermediate. The rate-limiting steps of multicopper oxidases, such as bilirubin oxidase and laccase, have not been firmly established. However, based on the $^{18}$O-KIEs determined in this study, and the comparison with the three enzymes with similar $^{18}$O-KIEs, a rate limiting copper-hydroperoxo formation by hydrogen atom abstraction seems likely. Similarly, the copper-dependent enzymes that displayed $^{18}$O-KIEs between 1.0086 and 1.011 are likely

characterized by a rate-limiting copper-superoxo formation, based on comparisons with $^{18}$O-EIEs (1.009 - 1.010). Accordingly, copper-superoxo formation has been suggested as the rate-limiting step for bovine serum amine oxidase and amine oxidase from *H. polymorpha* (Su and Klinman 1998; Mills et al. 2002). It can thus be assumed that tyrosinase, as well as L-ascorbate and diamine oxidase also have a rate-limiting step involving the formation of a copper-superoxo intermediate. For pea-seedling amine oxidase, for which a $^{18}$O-KIE of 1.014 ± 0.001 was determined (Mukherjee et al. 2008), a rate-limiting step involving

copper-peroxo formation has also been proposed. However, the preceding copper-superoxo formation is partially rate-limiting, which acts to lower the observed $^{18}$O-KIE value from the expected $^{18}$O-EIE range of 1.018-1.031 (Mukherjee et al. 2008).





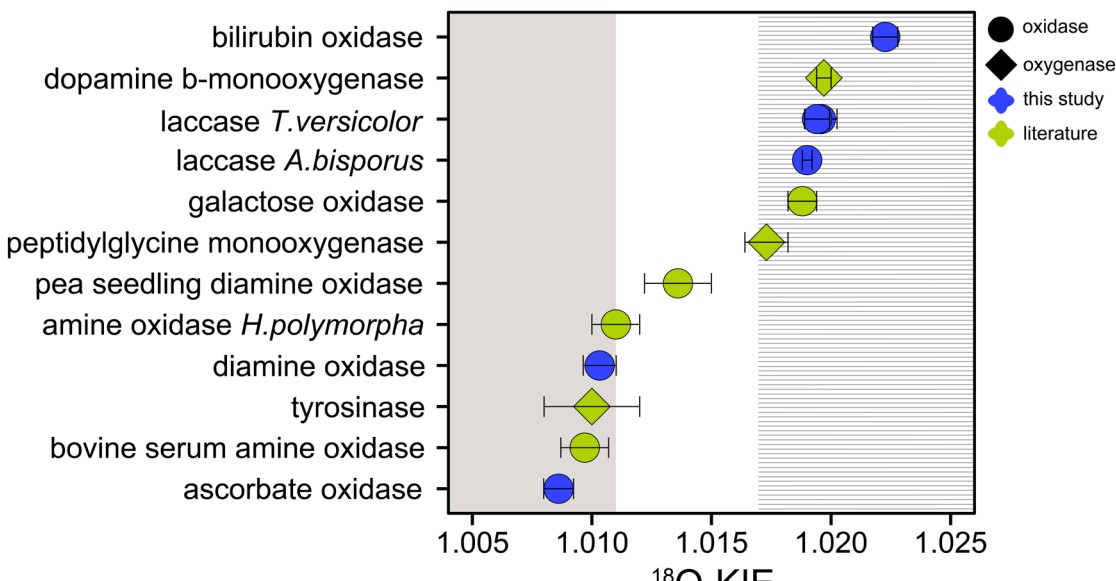

**Figure 5. $^{18}$O-KIEs for copper-dependent $O_2$-consuming oxidases (circles) and monooxygenases (diamonds) reported in this (blue) and previous studies (green) (Feldman et al., 1959; Tian et al., 1994; Su and Klinman, 1998; Francisco et al., 2003; Welford et al., 2007; Mukherjee et al., 2008; Humphreys et al., 2009). Error bars indicate 95 % confidence intervals or standard deviations. Grey and dashed areas represent expected $^{18}$O-KIE ranges for a rate-limiting copper-superoxo formation (grey area), and hydrogen atom abstraction by a copper-superoxo species (dashed area) (Mukherjee et al., 2008; Humphreys et al., 2009).**

The $^{18}$O-KIE of $1.0189 \pm 0.0005$ determined here for cytochrome-$c$ oxidase is consistent with previous reports from the literature (Ribas-Carbo et al. 1995; Cheah et al. 2014). Cytochrome-$c$ oxidase is a hetero di-nuclear copper-heme oxidase, in which a copper and a heme-iron are involved in the $O_2$-reduction mechanism (Yoshikawa and Shimada 2015). Iron-dependent enzymes form similar reactive oxygen intermediates, as described above for copper-dependent enzymes, including iron-superoxo (Fe(III)-OO•) and iron-hydroperoxo (Fe(III)-OOH) intermediates (see Fig. 6). In addition, iron can be oxidized further in certain active site structures to a high-valent iron-oxo (Fe(IV)=O) intermediate. The calculated $^{18}$O-EIEs are also similar in magnitude, with 1.008-1.009 for iron-superoxo formation, 1.014-1.017 for iron-hydroperoxo formation, and 1.029 for iron-oxo formation (Mirica et al. 2008). Previous studies have determined $^{18}$O-KIEs for 12 iron-dependent $O_2$-consuming enzymes showing a continuous range from $1.009 \pm 0.001$ for soybean lipoxygenase (Guy et al. 1992), to $1.0281 \pm 0.0004$ for alternative oxidase (Cheah et al. 2014). Observed $^{18}$O-KIEs for iron-dependent enzymes have consistently reflected the intrinsic $^{18}$O-KIE of the rate-limiting step, with increasing $^{18}$O-KIEs indicating a higher degree of $O_2$ reduction. For example, the $^{18}$O-KIE of soybean lipoxygenase (1.009-1.012), reflects a rate-limiting electron transfer to $O_2$ to form an iron-superoxo species (Guy et al. 1992; Knapp and Klinman 2003). The $^{18}$O-KIE of $1.015 \pm 0.001$ determined for hydroxyethyl phosphonate dioxygenase reflects a rate-limiting iron-hydroperoxo formation by hydrogen atom abstraction (Zhu et al. 2015). Finally, the $^{18}$O-KIE of 1-aminocyclopropyl-1-carboxylic acid oxidase ($1.0215 \pm 0.005$) reflects a rate-limiting iron-oxo formation (Mirica et al. 2008). For cytochrome-$c$ oxidase, a rate-limiting hydrogen atom abstraction by an iron-bound superoxo species with





concomitant O-O bond cleavage and formation of a high-valent iron-oxo intermediate has been suggested (Yoshikawa and

Shimada 2015). The corresponding $^{18}$O-KIE of $1.0189 \pm 0.0005$ determined in this study is in agreement with both a hydrogen

atom abstraction by a metal-superoxo species, as seen for many of the copper-dependent enzymes, as well as with the formation

of a high-valent iron-oxo species as described for 1-aminocyclopropyl-1-carboxylic acid oxidase.

**Figure 6. Simplified scheme of $O_2$ reduction steps performed by copper- and iron-dependent oxidases and oxygenases shown without interactions with (co-)substrates. M(0) indicates a metal ion in its most reduced state, which is typically Cu(I) or Fe(II), thus M(I)**
**corresponds to either Cu(II) or Fe(III). M(II)=O only occurs in iron-dependent enzymes as a high-valent iron-oxo species (Fe(IV)=O).**

## 5 Conclusions

The combined analysis of $^{18}$O-KIEs of $O_2$-consuming enzymes, determined in this and previous studies, enabled a

comprehensive evaluation of the variability of kinetic isotope effects within and between different active site structures, as

illustrated in Fig. 7. Notably, iron- and copper-dependent $O_2$-consuming enzymes displayed a narrower range of $^{18}$O-KIEs with

lower magnitudes (1.009 - 1.028) compared to flavin-dependent enzymes (1.020 - 1.058). This variability likely reflects

differences in electron transfer mechanisms, specifically inner- versus outer-sphere electron transfer. Within the flavin-

dependent $O_2$-consuming enzymes, the two distinct ranges of $^{18}$O-KIEs likely correspond to two different $O_2$ reduction

mechanisms, as discussed in section 4.1. Specifically, flavin-dependent enzymes with $^{18}$O-KIEs below 1.035 are likely

associated with a rate-limiting $O_2^{\bullet-}$ or FLOO$^-$ formation prior to FLOOH formation, potentially influenced by a rate-

contributing $O_2$ binding step that masks the intrinsic $^{18}$O-KIE. Conversely, flavin-dependent enzymes with $^{18}$O-KIEs above

1.04 are suggested to follow the alternative $O_2$ reduction pathway, in which $H_2O_2$ and oxidized flavin are formed directly from

FLH$^{\bullet}$ and $O_2^{\bullet-}$ without the formation of FLOOH. Similarly, the copper-dependent $O_2$-consuming enzymes investigated in this

and previous studies can be assigned to one of two groups (see Fig. 5). Enzymes with $^{18}$O-KIEs between 1.009 and 1.011 are

likely characterized by a rate-limiting copper-superoxo formation. Enzymes with $^{18}$O-KIEs between 1.017 and 1.022 are

suggested to have a rate-limiting hydrogen atom abstraction leading to the formation of a copper-hydroperoxo species. Based

on comparisons with calculated $^{18}$O-EIEs, a rate-limiting copper-peroxo species formation for the second group remains



possible, however, existing experimental evidence favors a copper-hydroperoxo formation. The continuous increase in $^{18}$O-KIEs observed for 13 iron-dependent $O_2$-consuming enzymes, including cytochrome-$c$ oxidase, reflects an increase in the

extent of $O_2$ reduction during the rate-limiting step, aligning with increasing $^{18}$O-EIEs calculated for metal-bound reactive oxygen intermediates. Consequently, if a $^{18}$O-KIE is determined for an unknown $O_2$-consuming enzymatic reaction, it appears that a value above 1.025 will typically be indicative of a flavin-dependent enzyme, whereas a value above 1.04 is characteristic for a flavin-dependent oxidase. By contrast, a $^{18}$O-KIE below 1.015 can be confidently assigned to a metal-dependent enzyme. However, distinguishing between copper- and iron-dependent enzymes within this range is not possible. Overall, the patterns

of isotopic fractionation of $O_2$ identified in this study can help clarify $O_2$ reduction mechanisms in other $O_2$-consuming enzymes. Furthermore, the improved understanding of the variability in isotopic fractionation of $O_2$ at the enzyme level can assist in the interpretation of the variability in isotope fractionation of $O_2$ observed at the organism or ecosystem levels. For instance, the trends observed for copper-dependent $O_2$-consuming enzymes may support the investigation of metabolic pathways carried out by environmentally relevant bacteria that possess copper-dependent $O_2$-consuming enzymes, such as

ammonia and methane monooxygenase. To further validate and support these findings, determining $^{18}$O-KIEs of additional flavin-dependent monooxygenases, and copper-dependent $O_2$-consuming enzymes in particular, would be highly valuable.



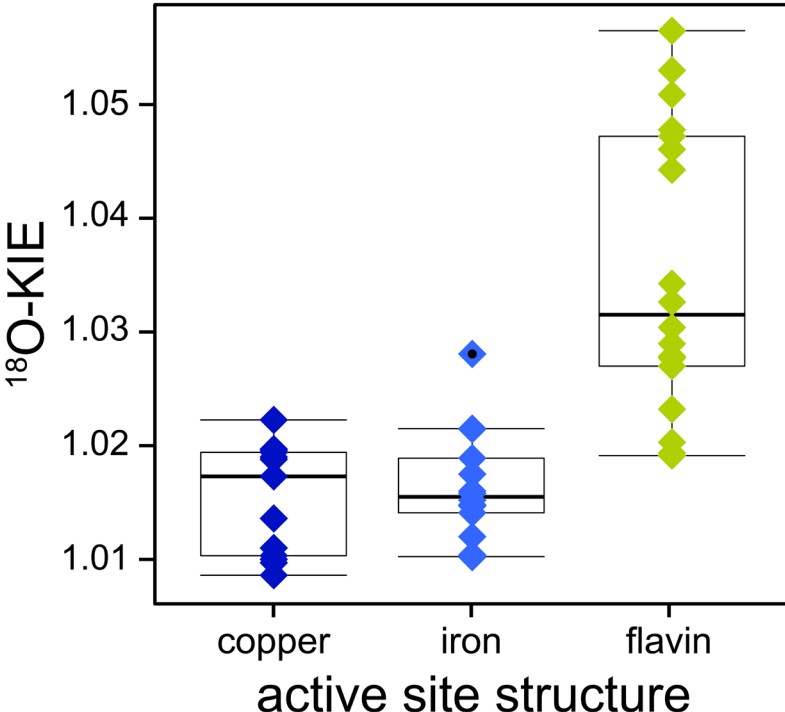

**Figure 7. $^{18}$O-KIEs of copper- (dark blue diamonds), iron- (light blue diamonds), and flavin-dependent (green diamonds) O$_2$-consuming enzymes obtained in this and previous studies. A list of literature values including references can be found in Table S1 in the supplement. Boxes represent interquartile range and median values. The whiskers extend to observations that fall within 1.5 times above or below the box size; individual points with black dots represent observations that fall out of this range.**

**Appendix A: Experimental conditions by enzyme**

All experiments were performed at room temperature (23 ± 1 °C) with an initial O$_2$ concentration of 270 ± 10 μM, unless stated otherwise. Typically, 6-8 experiments were performed to determine the $K_m$(S) values with constant conditions, except for initial organic substrate concentrations. The $K_m$(O$_2$) values were determined in single experiments at saturating substrate concentrations, unless noted otherwise. $^{18}$O-KIEs were determined with duplicate or triplicate experiments at saturating substrate concentrations.

**Alcohol oxidase**

Experiments with 0.4-32 mg protein L$^{-1}$ alcohol oxidase were performed in a 50 mM phosphate buffer (pH 7.5). To calculate $K_m$(S) values, initial O$_2$ consumption rates were determined at 8 different initial methanol concentrations from 0.5 to 5 mM and at 8 different initial ethanol concentrations from 0.5 to 200 mM. Product inhibition was tested separately with 1 mM formaldehyde, 1 mM acetaldehyde, and 1 mM H$_2$O$_2$. $K_m$(O$_2$) values were determined with 10 mM methanol and 200 mM





ethanol, respectively, at initial $O_2$ concentrations of $1200 \pm 100$ µM. Experiments with 2.5 mM methanol or 200 mM ethanol were performed to determine $^{18}$O-KIEs. Eq. A1 shows the reaction catalyzed by alcohol oxidase.

$$RCH_2OH + O_2 \rightarrow RCHO + H_2O_2 \qquad\qquad (A1)$$

**L-ascorbate oxidase**

Experiments with 0.06-0.19 mg protein $L^{-1}$ L-ascorbate oxidase were performed in a 50 mM acetate buffer (pH 5.0). To
calculate $K_m(S)$, initial $O_2$ consumption rates were determined at 8 different initial L-ascorbic acid concentrations from 0.06 to 3 mM. Concentrations of L-ascorbic acid above 3 mM resulted in inhibition of enzymatic activity. Product inhibition was tested with a reaction solution after complete consumption of 0.27 mM L-ascorbic acid. The $K_m(O_2)$ value and $^{18}$O-KIE were determined with 2.5 mM and 2 mM L-ascorbic acid, respectively. Eq. A2 shows the reaction catalyzed by L-ascorbate oxidase.

$$4\ L\text{-}ascorbate + O_2 \rightarrow 4\ monodehydroascorbate + 2\ H_2O \qquad\qquad (A2)$$

**Bilirubin oxidase**

Experiments with 0.7-2.5 mg protein $L^{-1}$ bilirubin oxidase were performed in a 100 mM Tris-HCl buffer (pH 8.5). To calculate $K_m(S)$, initial $O_2$ consumption rates were determined at 8 different initial bilirubin concentrations from 0.025 to 1 mM. Product inhibition was tested with a reaction solution after complete consumption of 0.3 mM bilirubin. The $K_m(O_2)$ value and $^{18}$O-KIE were determined with 1 mM bilirubin. Eq. A3 shows the reaction catalyzed by bilirubin oxidase.

$$2\ bilirubin + O_2 \rightarrow 2\ biliverdin + 2\ H_2O \qquad\qquad (A3)$$

**Cholesterol oxidase**

Experiments with 1.3-11 mg protein $L^{-1}$ cholesterol oxidase were performed in a 50 mM phosphate buffer (pH 7.5) with 1% (v/v) Thesit® and 10 % (v/v) isopropanol.  To calculate $K_m(S)$, initial $O_2$ consumption rates were determined at 6 different initial cholesterol concentrations from 0.1 to 1 mM. Product inhibition was tested separately with 0.3 and 1 mM $H_2O_2$ and with
a reaction solution after complete consumption of 0.3 mM cholesterol. The $K_m(O_2)$ value and $^{18}$O-KIE were determined with 1.5 mM cholesterol. Eq. A4 shows the reaction catalyzed by cholesterol oxidase.

$$cholesterol + O_2 \rightarrow cholest\text{-}5\text{-}en\text{-}3\text{-}one + H_2O_2 \qquad\qquad (A4)$$

**Choline oxidase**

Experiments with 3-10 mg/L choline oxidase were performed in a 50 mM phosphate buffer (pH 7.5). To calculate $K_m(S)$,
initial $O_2$ consumption rates were determined at 8 different initial choline concentrations from 0.075 to 4.5 mM. Product





inhibition was tested separately with 0.3 mM $H_2O_2$ and 0.3 mM betaine. The $K_m(O_2)$ value and $^{18}$O-KIE were determined with 10 mM and 2.5 mM choline, respectively. Eq. A5 shows the reaction catalyzed by choline oxidase.

$$\text{choline} + 2\,O_2 \rightarrow \text{betaine aldehyde} + O_2 + H_2O_2 \rightarrow \text{betaine} + 2\,H_2O_2 \tag{A5}$$

**Cytochrome-*c* oxidase**

Experiments with 1.5-2.3 mg protein L$^{-1}$ cytochrome-*c* oxidase were performed in a 10 mM phosphate buffer (pH 7.5) with 50 mM NaCl. $K_m(S)$ was not determined. Product inhibition was not tested. Experiments to determine $K_m(O_2)$ and $^{18}$O-KIE were performed with 25 μM cytochrome *c* and 3 mM ascorbic acid. Ascorbic acid was used to recycle the substrate by abiotically reducing ferricytochrome *c* to ferrocytochrome *c*. Eq. A6 shows the reaction catalyzed by cytochrome-*c* oxidase.

$$4\,\text{ferrocytochrome } c + O_2 + 4\,H^+ \rightarrow 4\,\text{ferricytochrome } c + 2\,H_2O \tag{A6}$$

**D-amino-acid oxidase**

Experiments with 1.8-5.9 mg protein L$^{-1}$ D-amino-acid oxidase were performed in a 50 mM Tris-HCl buffer (pH 8.2). To calculate $K_m(S)$, initial $O_2$ consumption rates were determined at 8 different initial D-alanine concentrations from 0.3 to 20 mM. Product inhibition was tested separately with 0.3 mM $H_2O_2$ as well as with 0.27 mM ammonium and 0.27 mM pyruvate. The $K_m(O_2)$ value and $^{18}$O-KIE were determined with 20 mM D-alanine. Eq. A7 shows the reaction catalyzed by D-amino-635 acid oxidase.

$$\text{D-alanine} + H_2O + O_2 \rightarrow \text{pyruvate} + NH_4^+ + H_2O_2 \tag{A7}$$

**Diamine oxidase**

Experiments with 800-5000 mg/L diamine oxidase were performed in a 50 mM phosphate buffer (pH 7.2). To calculate $K_m(S)$, initial $O_2$ consumption rates were determined at 6 different initial histamine concentrations from 0.025 to 0.5 mM.
Concentrations of histamine above 0.5 mM resulted in inhibition of enzymatic activity. Product inhibition was tested with a reaction solution after complete consumption of 0.25 mM histamine. The $K_m(O_2)$ value and $^{18}$O-KIE were determined with 0.4 mM histamine. Eq. A8 shows the reaction catalyzed by diamine oxidase. The enzyme provided by the manufacturer was tested positively for catalase activity. Thus, the $H_2O_2$ formed during the reaction of histamine with diamine oxidase was immediately converted to $O_2$ and $H_2O$ (see section 3.3 for implications of $O_2$ formation on $^{18}$O-KIE determination).

$$\text{histamine} + H_2O + O_2 \rightarrow \text{(imidazol-4-yl)acetaldehyde} + NH_3 + H_2O_2 \tag{A8}$$



**Glucose oxidase**

Experiments with 9-41 mg protein $L^{-1}$ glucose oxidase were performed in a 100 mM acetate buffer (pH 5.0). To calculate $K_m(S)$ values, initial $O_2$ consumption rates were determined at 7 different initial D-glucose concentrations from 0.45 to 70 mM and at 11 different initial D-mannose concentrations from 0.45 to 100 mM. Product inhibition was tested separately with 0.3 mM $H_2O_2$ and with reaction solutions after complete consumption of 0.45 mM D-mannose and 0.27 mM D-glucose, respectively. The $K_m(O_2)$ values were determined with 40 mM D-glucose and 100 mM D-mannose, respectively. The $^{18}$O-KIEs were determined with 40 mM D-glucose or 40 mM D-mannose. Eq. A9 shows the reaction catalyzed by glucose oxidase with D-glucose.

$$\beta\text{-}D\text{-}glucose + O_2 \rightarrow D\text{-}glucono\text{-}1,5\text{-}lactone + H_2O_2 \tag{A9}$$

**Kynurenine 3-monooxygenase**

Experiments with 3-9 mg/L kynurenine 3-monooxygenase (KMO) were performed in a 20 mM HEPES buffer (pH 7.5). $K_m(S)$ was not determined. Product inhibition was tested with a reaction solution after complete consumption of 0.3 mM L-kynurenine. To calculate $K_m(O_2)$, initial $O_2$ consumption rates were determined with 1 mM L-kynurenine, 0.5 mM NADPH and 2 mM dithiothreitol at 8 different initial $O_2$ concentrations from 25 to 260 µM. The $^{18}$O-KIE was determined with 1 mM L-kynurenine, 0.5 mM NADPH and 2 mM dithiothreitol. Eq. A10 shows the reaction catalyzed by KMO.

$$L\text{-}kynurenine + NADPH + H^+ + O_2 \rightarrow 3\text{-}hydroxy\text{-}L\text{-}kynurenine + NADP^+ + H_2O \tag{A10}$$

**Laccase from *Agaricus bisporus***

Experiments with 10-100 mg/L laccase from *Agaricus bisporus* were performed in a 50 mM acetate buffer (pH 5.5). To calculate $K_m(S)$, initial $O_2$ consumption rates were determined at 10 different initial hydroquinone concentrations from 0.05 to 20 mM. Product inhibition was tested with 0.54 mM *p*-benzoquinone. The $K_m(O_2)$ value and $^{18}$O-KIE were determined with 15 mM hydroquinone. Eq. A11 shows the reaction catalyzed by laccase with hydroquinone.

$$2 \, hydroquinone + O_2 \rightarrow 2 \, p\text{-}benzoquinone + 2 \, H_2O \tag{A11}$$

**Laccase from *Trametes versicolor***

Experiments with 10-100 mg/L laccase from *Trametes versicolor* were performed in a 50 mM acetate buffer (pH 5.5). To calculate $K_m(S)$ values, initial $O_2$ consumption rates were determined at 10 different initial hydroquinone concentrations from 0.005 to 15 mM and at 7 different initial ABTS concentrations from 0.06 to 7.5 mM. Product inhibition was tested with 0.54 mM *p*-benzoquinone and with a reaction solution after complete consumption of 1.2 mM ABTS. The $K_m(O_2)$ values were determined from a single experiment with 15 mM hydroquinone and from initial $O_2$ consumption rates with 3.8 mM ABTS



and 6 different initial $O_2$ concentrations from 25 to 265 µM. The $^{18}O$-KIEs were determined with 7.5 mM hydroquinone and
4 mM ABTS, respectively. Eq. A12 shows the reaction catalyzed by laccase with ABTS.

$$4\,ABTS^{2-} + 4\,H^+ + O_2\; \rightarrow 4\,ABTS^{-\bullet} + 2\,H_2O \tag{A12}$$

**L-lactate oxidase**

To calculate $K_m(S)$, initial $O_2$ consumption rates were determined at six different initial L-lactic acid concentrations from 0.1
to 10 mM in a 50 mM phosphate buffer (pH 7.0) with 20 mM KCl and 2.3 mg/L enzyme. Product inhibition was tested
separately with 0.3 mM pyruvate and 0.3 mM $H_2O_2$. The $K_m(O_2)$ value and $^{18}O$-KIE were determined in a 50 mM HEPES
buffer (pH 7.0) with 50 mM KCl containing either 10 mM L-lactic acid and 2.3 mg/L enzyme or 5 mM L-lactic acid and 1.2
mg/L enzyme. Eq. A13 shows the reaction catalyzed by L-lactate oxidase.

$$L\text{-}lactate +\; O_2 \rightarrow pyruvate\; +\; H_2O_2 \tag{A13}$$

**L-lysine oxidase**

Experiments with 0.3-2.2 mg protein $L^{-1}$ L-lysine oxidase were performed in a 50 mM phosphate buffer (pH 8.0). To calculate
$K_m(S)$, initial $O_2$ consumption rates were determined at 6 different initial L-lysine concentrations from 0.01 to 2 mM. Product
inhibition was tested separately with 0.3 mM $H_2O_2$ and with a reaction solution after complete consumption of 0.3 mM L-
lysine. The $K_m(O_2)$ value and $^{18}O$-KIE were determined with 2.3 mM and 2 mM L-lysine, respectively. Eq. A14 shows the
reaction catalyzed by L-lysine oxidase.

$$L\text{-}lysine + H_2O +\; O_2 \rightarrow 6-amino\text{-}2\text{-}oxohexanoate\; + NH_3 +\; H_2O_2 \tag{A14}$$

**Pyruvate oxidase**

Experiments with 0.3-1.3 mg protein $L^{-1}$ pyruvate oxidase were performed in a 50 mM phosphate buffer (pH 6.7) with 1 mM
thiamine diphosphate, 1 mM $MnSO_4$ and 10 µM FAD. $K_m(S)$ was not determined. Product inhibition was tested separately
with 0.27 mM sodium bicarbonate and 0.27 mM $H_2O_2$. The $K_m(O_2)$ value was determined with 100 mM pyruvate. The $^{18}O$-
KIE was determined with 25, 50, and 100 mM pyruvate. Prior to starting an experiment, pyruvate oxidase was incubated with
1 mM thiamine diphosphate, 1 mM $MnSO_4$, 10 µM FAD, and 5-100 mM pyruvate for 10 minutes at room temperature. Eq.
A15 shows the reaction catalyzed by pyruvate oxidase.

$$pyruvate + phosphate +\; O_2 \rightarrow acetyl\ phosphate + CO_2 +\; H_2O_2 \tag{A15}$$



**Sarcosine oxidase**

Experiments with 0.5-10 mg/L sarcosine oxidase were performed in a 100 mM Tris-HCl buffer (pH 8.3). To calculate $K_m(S)$ values, initial $O_2$ consumption rates were determined at 6 different initial sarcosine concentrations from 5 to 100 mM. Product inhibition was tested separately with 0.3 mM glycine, 1 mM formaldehyde, and 0.3 mM $H_2O_2$. The $K_m(O_2)$ value and [18]O-KIE were determined with 100 mM and 50 mM sarcosine, respectively. Eq. A16 shows the reaction catalyzed by sarcosine oxidase.

$$\text{sarcosine} + H_2O + O_2 \rightarrow \text{glycine} + \text{formaldehyde} + H_2O_2 \tag{A16}$$

**Appendix B: Enzyme assays for [18]O-kinetic isotope effects in $O_2$-purged buffer**

Alcohol, choline, and L-lysine oxidase exhibited $K_m(O_2)$ values above air-saturation. For this reason, in addition to the enzyme assays described in section 2.2, the [18]O-kinetic isotope effects ([18]O-KIEs) of these enzymes were additionally performed in $O_2$-purged buffer solutions under otherwise identical experimental conditions (see Appendix A). Enzyme assays with alcohol
and choline oxidase were performed each in 12 identically filled crimp-top vials as described in section 2.2. Enzyme assays with L-lysine oxidase were performed directly in 8 Exetainers that were sacrificed at different time-points. Exetainers were filled completely with assay solution and closed, before a small volume of enzyme or substrate solution was injected through the septa to initiate the reaction. Prior to sampling, the remaining $O_2$ concentration was measured with a fiber-optic oxygen microsensor. After measuring $O_2$ concentrations, the reaction was stopped by injecting 200 μL of a 3 M HCl solution through
the septa with a gas-tight glass syringe, while simultaneously piercing the septa with a small exhaust needle. After enzyme injection, before measuring $O_2$ concentrations, and after HCl additions, Exetainers were shaken vigorously. To create a He headspace in the Exetainer, 5 mL assay solution was removed with a 10 mL gas-tight glass while the Exetainer was connected to a slow stream of He gas. Procedural blanks were prepared by completely filling Exetainers with $N_2$-purged water in an anaerobic glove box with a $N_2$ atmosphere (GS GLOVEBOX Systemtechnik, residual $O_2$ content < 1 ppm). Under ambient
atmosphere, 200 μL NaOH were then injected through the septa into the closed Exetainer. Control samples and quantification standards were prepared by completely filling Exetainers with leftover assay solution without enzyme or with air-equilibrated water, respectively. For blanks, control samples, and quantification standards, a 5 mL He headspace was created as described for the assay samples. The resulting [18]O-KIEs were $1.029 \pm 0.004$ for alcohol oxidase, $1.019 \pm 0.002$ for choline oxidase, and $1.04 \pm 0.01$ for L-lysine oxidase.



## Appendix C: Substrate-to-O$_2$ consumption stoichiometries of laccase

With laccase from *T. versicolor*, the substrate-to-O$_2$ consumption stoichiometry was determined for the two substrates hydroquinone and ABTS. Enzyme assays were performed in air-saturated buffer as described in section 2.1, but with a limiting amount of substrate. O$_2$ concentrations were stable before substrate addition. After the addition of 50 µM hydroquinone, O$_2$ concentrations decreased rapidly from 275 µM to 249 µM and remained stable thereafter. Assuming all hydroquinone was consumed, this decrease in O$_2$ concentration corresponds to a substrate-to-O$_2$ consumption stoichiometry of 1.92 to 1. After the addition of 51 µM ABTS, O$_2$ concentrations decreased rapidly from 262 µM to 249 µM and slowly thereafter. It is likely that the initial fast decrease in O$_2$ concentration is the result of the enzymatic reaction catalyzed by laccase, while the later slower O$_2$ consumption is a result of abiotic reaction between the radical product ABTS$^{\cdot -}$ and O$_2$. Assuming all ABTS was consumed in the initial fast reaction, the substrate-to-O$_2$ consumption stoichiometry was 4.0 to 1.

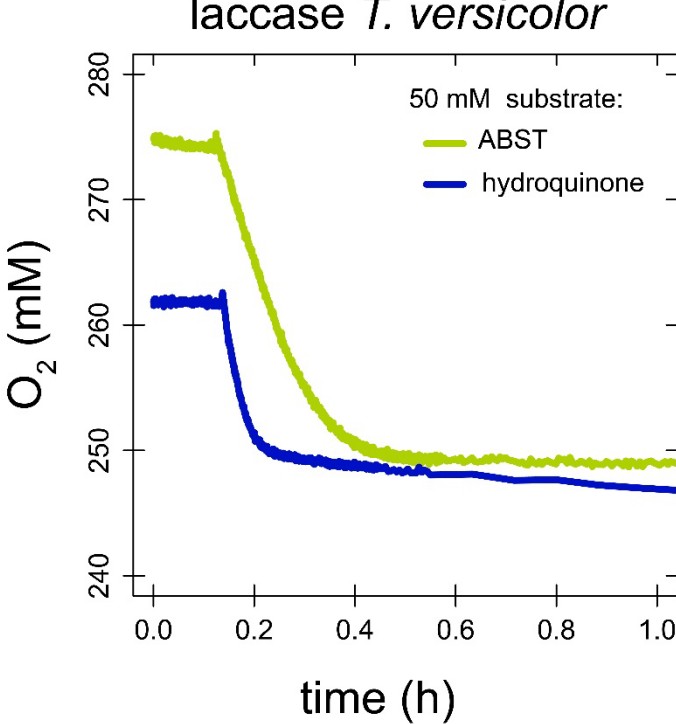


**Figure C1. O$_2$ consumption profiles by laccase from *T. versicolor* when supplied with 50 µM ABTS (green line) or hydroquinone (blue line).**

**Data availability**

All data presented in this study are available at https://doi.org/10.5281/zenodo.14765061.



**Author contribution**

CFMC contributed to conceptualization, investigation, methodology, visualization, writing (original draft, review and editing). MFL contributed to supervision, writing (review and editing). SGP contributed to conceptualization, funding acquisition, methodology, supervision, writing (review and editing).

**Competing interests**

The authors declare that they have no conflict of interest.

**Acknowledgments**

We thank Thomas Kuhn for his support with IRMS measurements.

**Financial support**

This work was supported by the Swiss National Science Foundation (Grant no. PZ00P2_186083).

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
