# Peer review of "Variability in oxygen isotopic fractionation of enzymatic $O_2$ consumption"

_EGUsphere, 2025_

## Author Comment (AC2)

**Appendix D: Derivation of apparent correlation between $^{18}$O-KIE and $K_m(O_2)$**

To reconcile the apparent correlation between $^{18}$O-KIEs and $K_m(O_2)$ as shown in Fig. 4, we consider a simple two-step enzymatic reaction involving a reversible $O_2$ binding step and an irreversible reaction step converting enzyme-bound $O_2$ into products (either $H_2O_2$ or hydroxylated organic substrate) as shown in Eq. D1,

$$O_2 \underset{k_2}{\overset{k_1}{\leftrightarrow}} E\text{-}O_2 \overset{k_3}{\rightarrow} P \tag{D1}$$

where E-$O_2$ is the enzyme-bound $O_2$, P represents the reaction products, and $k_1$, $k_2$, and $k_3$ are elementary reaction rate constants of the forward and backward reactions. In this case, the measured $^{18}$O-KIE is related to the intrinsic equilibrium and kinetic isotope effects of the two elementary steps through the forward commitment to catalysis, $k_3/k_2$, as shown in Eq. D2 (Cleland 2005),

$$^{18}\text{O-KIE}_{measured} = \frac{EIE_1 KIE_3 + {k_3}/{k_2} KIE_1}{1 + {k_3}/{k_2}} \tag{D2}$$

where $EIE_1$ is the equilibrium isotope effect of the $O_2$ binding step and $KIE_1$ and $KIE_3$ are the kinetic isotope effects of the $O_2$ binding and reaction steps associated with rates $k_1$ and $k_3$, respectively. From Eq. D2, two extreme cases can be derived. If $O_2$ binding alone is rate-limiting ($k_3 >> k_2$), the measured $^{18}$O-KIE will approximate $KIE_1$. If the second reaction step is rate-limiting ($k_3 << k_2$), the measured $^{18}$O-KIE will approximate the product of $EIE_1$ and $KIE_3$. When we start with the latter case, which has a small $k_3/k_2$, and increase the forward commitment gradually, the measured $^{18}$O-KIE will slowly decrease assuming $KIE_1$ is smaller than $EIE_1 KIE_3$. For such a reaction, plotting measured $^{18}$O-KIEs vs. $k_3/k_2$ will yield a similar (apparently linear) trend as shown in Fig. 4 as long as the commitment factor ($k_3/k_2$) is below 1. As shown in Eq. D3, $k_3/k_2$ can be related to $K_m(O_2)$, if we consider $K_m(O_2)$ to be $(k_3 + k_2)/k_1$ and $K_d$, the dissociation constant of $O_2$, to be $k_2/k_1$. The trend observed in Fig. 4 can thus be explained if $K_d$ varies much less than $K_m(O_2)$ for this set of enzymes.

$$\frac{k_3}{k_2} = \frac{k_3}{k_2} + \frac{k_2}{k_2} - 1 = \frac{k_3 + k_2}{k_2} - 1 = \frac{K_m(O_2)}{K_d} - 1 \tag{D3}$$

---

## Author Response (AR1)

**Author's response to referee comments**

Comments by the referee are marked as "**Ref {comment no.}**", answers by the authors are given as "**Auth {comment no.}**". The text extracts following author comments show the revised parts of the manuscript with **line numbers** corresponding to the **Author's track-changes file.**

**Responses to referee 1**

**Ref 1.0** The manuscript describes the isotope effects of O2-consuming enzymes and will be useful for many disciplines that seek to investigate respiration processes in the environment. The authors have tested a wide range of organic substrates and corresponding enzymes, extending the dataset in the literature, and have provided useful conjecture on mechanisms that may determine isotope distributions. The isotope methodology is modern, eloquent, and explained in detail. I lack the expertise to provide a thorough review of the experimental set-up and concentrations chosen to perform the enzyme assays; however, the text explaining these choices is transparent and polished, and for my understanding, sufficient information has been provided in the Appendices and available data sets. I recommend the manuscript for publication after minor revision.

**Auth 1.0** We thank the referee for their overall positive evaluation and specific feedback.

**Ref 1.1** Eq 4 is given for the kinetic isotope effect (Line 78). It appears to be incorrect as one cannot obtain the reported values of ~1.010 to 1.050 given the reported epsilon values. The KIE (traditionally alpha) should be = (eps/1000) +1, or eps = (alpha – 1) * 1000.

Please be consistent with reporting eps and KIE values. This will likely require 1-2 sentences clarifying the definition of these parameters describing the isotope effect. Presumably, KIE values > 1 should have a positive eps value, whereas KIE < 1 have a negative eps value, after the equation above. In the current version of the manuscript, all reported KIE values in the current study are > 1, mostly negative eps values are reported in the Introduction (Lines 44, 111-114, etc.). This is likely due to reversal of the connotation, with heavy/light ratios of either the reaction substrates or products being in the numerator or denominator. In other words, whereas eps is reported from the perspective of the product of the reaction (negative value connotates that the product is depleted in the heavy isotope relative to substrate), the KIE values (i.e., alpha) are reported from the perspective of the substrate of the reaction, which becomes relatively enriched in the heavy

isotope. For clarity, it would be useful to also report the eps values for the enzymes tested (in Table 2, if possible), consistent with literature cited in these Introduction lines.

**Auth 1.1** Indeed, the definition of isotope effects can vary between different disciplines. We follow recommendations by Coplen (2011) as indicated in line 37 and define the kinetic isotope effect (KIE) in eq. 3, as the ratio of rate constants for the reaction of light vs. heavy isotopologues of $O_2$. This definition is in accordance with KIE values reported for $O_2$-consuming enzymes as referenced throughout the manuscript. The referee associated the term "isotope effect" with α (alpha), which is referred to as isotopic fractionation factor in Coplen (2011). We now include the inverse relationship between α and KIE, as defined above, in Eq. (4). As recommended by Coplen (2011) we refrain from adding the factor 1000, but otherwise we now display the same equation as given by the referee (α = ε+1). To improve clarity, we have included the fact that ε values are typically reported in permil in line 37. In summary, KIEs > 1, which is the case for all reported KIEs for $O_2$ consumption reactions, result in α values < 1 and negative ε values as reported in lines 44 and 114. As requested by the referee, we have now included ε values for the enzymes tested in Table 2 so that all our conversions are transparent and comparisons with different literature values are facilitated.

**Lines 37-38:** "... isotopic fractionation ... can be quantified with, for example, $^{18}\varepsilon$ values (see Eq. (1)), which are typically reported in permil (‰) (Coplen 2011):"

**Lines 78-80:** "Apparent $^{18}O$-KIEs are related to $^{18}\varepsilon$ and $^{18}\alpha$ values as shown in Eq. (4).

$$^{18}O\text{-}KIE = \left(^{18}\alpha\right)^{-1} = (^{18}\varepsilon + 1)^{-1}"$$

**Ref 1.2** Line 260 – How were the concentrations of organic substrate measured, as implied by this sentence? If only O2 concentrations were measured, please add text to clarify. It's not clear if this is what is explained in Line 263-264. Presumably the initial substrate concentrations are assumed from experimental preparation and concentrations were not measured over time.

**Auth 1.2** That is correct. We used the initial added organic substrate concentration to calculate $K_m(S)$ as commonly done in enzyme kinetic studies. To clarify this, the manuscript text has been changed as indicated below.

**Line 261:** "... $[i]_t$ is the initial (t=0), nominal concentration of an organic substrate (S) or the measured concentration of $O_2$ at time t, ..."

**Ref 1.3.** I suggest to improve the reaction mechanisms and Appendix equations, by better depicting the distribution of $O_2$ in the products of the reaction (see below). The appendix equations could be similarly color-coded as Fig. 6, for example.

**Auth 1.3** The text in the appendix has been changed by colour coding O distribution in the products as suggested by the referee.

**Ref 1.4** The discussion/conclusion justly describes how the findings of this study may be applied to delineate mechanisms of oxic, enzymatic respiration. It could be enhanced with discussion of other processes that presumably influence d18O of not only oxygen gas but also oxygen in oxidized, molecular end-products (e.g., D glucono- 1,5- lactone, Line 654; benzoquinone, Line 667; etc.). For example, I would appreciate to if the findings were discussed in the context of known isotope effects of biosynthesis (i.e., the reverse reaction of respiration).

**Auth 1.4** As requested, we have extended our conclusion section by including a discussion on the O-isotopic composition of reaction products (see below). In the context of $O_2$-consuming enzymes, we consider two groups of products most relevant, namely O-containing aromatic compounds and $H_2O_2$. D-glucono-1,5-lactone, as suggested by the referee, does not incorporate oxygen atoms from $O_2$ during the glucose oxidase reaction. Its $\delta^{18}O$ reflects the isotopic composition of the original glucose precursor and water molecules from earlier biosynthetic steps, rather than any fractionation associated with $O_2$ reduction. While hydroquinone, the precursor of benzoquinone, can be formed by oxygenase enzymes, we are not aware of any reported measurements of O-isotopic composition of hydroquinone or benzoquinone. We have thus not included this example specifically but rather discuss O-containing aromatic compounds in general.

**Lines 586 ff:** "... is not possible. In contrast to the differences observed for different active site structures, the ranges of $^{18}O$-KIEs associated with oxygenases (1.009-1.030) and oxidases (1.010-1.057) overlap. Nevertheless, these ranges provide benchmarks for comparisons with the O-isotopic composition of the main products of these enzymes, namely O-containing aromatic compounds and $H_2O_2$, respectively. $\delta^{18}O$ values of natural, aromatic compounds, in which O-atoms primarily origin from $O_2$, have been measured to be 5-19 ‰ (Schmidt et al. 2001). Assuming a constant pool of dissolved $O_2$ with a $\delta^{18}O$ value of 24 ‰ suggests underlying $^{18}\varepsilon$ values for the biosynthesis of these compounds in the range of -5 to -19 ‰, which agrees well with the range of $^{18}\varepsilon$ values (-9 to -30 ‰) reported in this and previous studies for oxygenase enzymes. For $H_2O_2$, measurements of O-isotopic composition in natural waters are scarce. In

rainwater, $\delta^{18}O$ values of $H_2O_2$ were 22-53 ‰ (Savarino and Thiemens 1999). Consequently, $H_2O_2$ is more enriched in $^{18}O$ than expected from $^{18}\varepsilon$ values of oxidase reactions (-9 to -53 ‰). However, this is not surprising considering that $H_2O_2$ can also be formed through different processes and rapidly reacts further, which will likely lead to an increase in $\delta^{18}O$ values as observed. Overall, ..."

**Ref 1.5** Line 19 – change "which" to "associated with"

**Auth 1.5** The text has been changed accordingly.

**Lines 19-20:** "... displayed a narrower range of $^{18}O$-KIEs, with overall lower values (from 1.009 to 1.028), associated with an increase in the degree of ..."

**Ref 1.6** Line 114 – change 18O-e to 18eps, or the variable 18O-eps needs to be defined.

**Auth 1.6** We thank the referee for their attention to detail. The notation "$^{18}O$-$\varepsilon$" should indeed be "$^{18}\varepsilon$". The variable "$^{18}\varepsilon$" is now consistently used throughout the manuscript.

**Line 117:** "... values of -9 ‰ to -50 ‰, significantly exceeding the previously mentioned range of $^{18}$-$\varepsilon$ values observed for respiratory $O_2$ ..."

**Ref 1.7** Fig. 3 – What is the "S" that is reduced/oxidized? Could the oxidized form of S be in red font?

**Auth 1.7** To clarify, the final sentence of the figure caption has been changed as shown below. Because oxygen atoms from $O_2$ are not incorporated into the oxidized substrate ($S_{ox}$) during oxidase catalyzed reactions, the font color was not changed.

**Figure 3.** "... by oxidases. $S_{red.}$ and S-H represent an organic substrate before oxidation by an oxidase or monooxygenase, respectively, while $S_{ox}$ and S-O(H) represent the corresponding organic reaction products."

**Ref 1.8** Fig. 6 – The red text appears to track oxygen atoms originating from $O_2$ in the reaction. Should the O in $H_2O$ also be red?

**Auth 1.8** We appreciate this suggestion and have colored the oxygen atom in $H_2O$ in Fig. 6 red to consistently track oxygen atoms originating from $O_2$.

**Responses to referee 2**

**Ref 2.0** This is a very interesting paper that impacts both fundamental enzymology/ biochemistry and environmental sciences.

The extension of oxygen kinetic isotope effects (KIEs) to a wide scope of enzymes, as well as the availability of comparative values for $^{18}O$ and $^{17}O$ isotope effects introduces a comprehensive resource for researchers. The authors are to be congratulated.

The experimental measurements are carefully collected and for the most part meaningfully interpreted.

The Discussion however could be improved after consideration of each of the comments below. Response/ revision to address these issues is important, prior to acceptance for publication.

**Auth 2.0** We thank the referee for their overall positive evaluation and input for improving our discussion section.

**Ref 2.1** pp. 16-17. The interpretation of kinetic oxygen isotope effects rest on the fact that these are competitive measurements and therefore always reflect kcat/Km parameters. Thus, the $^{18}O$ KIE is reflective on all steps from $O_2$ binding up to and including the first irreversible step. If activation of O2 is multi-step, many steps can be reflected in the measurement. However, once an irreversible step has taken place during the $^{18}O$ measurement, the value will be independent of subsequent, kinetically significant steps. For this reason, there can (and often are) different rate limiting steps when reporting on kcat/Km vs kcat.

**Auth 2.1** We have specified the steps that are covered by measurements of $^{18}O$-KIEs more carefully in lines 74-75 and 400-403 (see below). Generally, we included comparisons with other studies that also report competitive $^{18}O$-KIEs and thus effects on kcat/Km for interpretations of reaction mechanisms relating to $O_2$ activation. The study we included in lines 404-405 was indeed an exception and upon closer examination, with the referee's comment in mind, does not constitute a contradiction as stated in the original submission. We have thus removed this sentence to avoid confusion.

**Lines 75-77:** "Experimentally determined $^{18}O$-KIEs reflect the O-isotopic fractionation occurring in all elementary reaction steps  up to, and including, the rate-limiting step (Roth 2007)."

**Lines 403-408:** "When comparing experimental $^{18}O$-KIEs to calculated $^{18}O$-EIEs, it is generally assumed that a measured $^{18}O$-KIE (i) reflects intrinsic $^{18}O$-KIEs of all electron and proton transfer

steps up to, and including, the rate-limiting (i.e., first irreversible) step and (ii) is similar to, but not larger than, the $^{18}$O-EIE calculated for the formation of the product/intermediate after the rate-limiting step (Roth and Klinman 2005; Roth 2007). Based on these $^{18}$O-EIEs, the reduction of $O_2$ by KMO is thus likely characterized by a rate-limiting $O_2^{•-}$ or FLOO$^-$ formation. However, this conclusion conflicts with studies suggesting that substrate hydroxylation is the rate-limiting step (Özkılıç and Tüzün 2019)."

**Ref 2.2** pp 16-17. The magnitude of a kinetic isotope effects has an additional component than the equilibrium isotope effects. This is because a KIE also contains a reaction coordinate frequency that can be altered (to some extent) by isotopic labelling {cf. Angeles-Boza, Chem Science 5, 1141 (2014)}.

**Auth 2.2** We have specified this difference between KIEs and EIEs in lines 80-82, where this comparison first comes up.

**Lines 82-85:** "Because $^{18}$O-KIEs contain an additional reaction coordinate frequency compared to $^{18}$O-EIEs, intrinsic $^{18}$O-KIEs cannot be difficult toeasily calculated, (Roth 2007). Therefore, calculated $^{18}$O-EIEs are often used as a reference to assign experimentally determined $^{18}$O-KIEs to a specific rate-limiting step (Roth and Klinman 2005)."

**Ref 2.3** p 18. It is difficult to make a direct comparison between kcat and kcat/Km because you are comparing rate constants with different units, s$^{-1}$ and M$^{-1}$s$^{-1}$, respectively. The best way to think about the impact of the affinity of O2 on the $^{18}$O KIE is through the expression:

kobs = k1k2/ (k-1 +k2)

where k1 is the binding rate constant, k-1 is the off rate constant and k2 is the chemical step. If the off rate is slow (tightly bound O2?) then the $^{18}$O will only reflect k1. If k-1 is fast, binding approximates an equilibrium situation and the kobs is Kdk2.

It is very curious and interesting that the largest values in Table 4 occur for the reactions with the smaller Km. This may be the result of a small k-1 combined with a rate limiting binding step that is accompanied by electron transfer

**Auth 2.3** In lines 430-455, we compared kcat and kcat/Km not quantitatively, but on a more conceptual basis, similar to the treatment in Northrop 1998. We understand, however, that this description can lead to misunderstandings and have revised this section based on the referee's comment above. Our revised manuscript also contains an additional appendix (D), providing

mathematical considerations for this section that we consider to be relevant only to expert readers.

**Lines 433 ff:** "For KMO, cholesterol, choline, and glycolate oxidase, as well as glucose oxidase with  3 different substrates, which we consider to share a common reaction mechanism, we found a tentative correlation between $^{18}$O-KIEs and the corresponding $K_m(O_2)$ values (see Fig. 4). The $K_m(O_2)$ values for glucose oxidase with the substrate 2-deoxy-D-glucose and for glycolate oxidase were reported to be 25 ± 5 µM and 210 µM, respectively (Macheroux et al. 1991; Roth and Klinman 2003). Based on the limited number of data points, we do not consider the correlation to be necessarily linear as shown in Fig. 4. However, the data clearly indicates that enzymes with lower $K_m(O_2)$ values have higher $^{18}$O-KIEs, ranging from choline oxidase with a $K_m(O_2)$ of 298 ± 20 µM and a $^{18}$O-KIE of 1.0194 ± 0.0006, to glucose oxidase with D-mannose as the substrate with a $K_m(O_2)$ of 3.9 ± 0.6 µM and a $^{18}$O-KIE of 1.0341 ± 0.0005. Since $^{18}$O-KIEs reflect the ratios of reaction rates of the different $O_2$ isotopologues, a correlation between $^{18}$O-KIE and $K_m(O_2)$ only makes sense when we consider the kinetic properties of the Michaelis constant.  (Northrop 1998).  In $O_2$-consuming enzymes, $O_2$ typically binds to the enzyme after binding of the organic substrate (oxygenases), or in a ping-pong mechanism (oxidases) (Malmstrom 1982; Romero et al. 2018). Thus, we can describe the consumption of $O_2$ by these enzymes kinetically with a two-step reaction, where $O_2$ first binds reversibly to the enzyme, followed by an irreversible reduction step of $O_2$. In such a case, the measured $^{18}$O-KIE depends on the intrinsic $^{18}$O-KIE and $^{18}$O-EIE of the $O_2$ binding step, the $^{18}$O-KIE of the irreversible reduction step, and the forward commitment to catalysis. This commitment factor is the ratio of two elementary reaction rates, namely the rate of the irreversible reduction step divided by the rate of the backward reaction of $O_2$ binding (see Appendix D for details). In fact, as long as the reduction step is slower than the backward binding step, and thus the commitment factor below 1, the measured $^{18}$O-KIE will show an apparently linear trend with an increasing commitment factor, similar to the trend observed in Fig. 4. For this set of enzymes, it thus appears that $K_m(O_2)$ is ~~the only step covered by "$v_{max}/K_m(O_2)$" but not by "$v_{max}$" is $O_2$ binding. Therefore, when $K_m(O_2)$ is very large, "$v_{max}/K_m(O_2)$" is much smaller than "$v_{max}$", and $O_2$ binding must be slower than the catalytic step. However, as $K_m(O_2)$ decreases, "$v_{max}/K_m(O_2)$" becomes closer to "$v_{max}$", and $O_2$ binding contributes less to the overall reaction rate. Consequently, $K_m(O_2)$ can be~~

interpreted as a proxy for the forward commitment to catalysis or the extent to which $O_2$ binding contributes to the overall reaction rate. One can indeed mathematically relate $K_m(O_2)$ to the commitment factor, as shown in Appendix D, and reconcile the observed decrease in $^{18}O$-KIE with increasing $K_m(O_2)$ values, if (i) $O_2$ binding and unbinding is faster than $O_2$ reduction for all enzymes but with different degrees of forward commitment, (ii) the intrinsic $^{18}O$-KIE for $O_2$ reduction is larger than for $O_2$ binding while all intrinsic isotope effects are close to identical for these enzymes, and (iii) the dissociation constant (the ratio of backward and forward reaction rates of $O_2$ binding) of these enzymes varies much less than $K_m(O_2)$. If  $O_2$ binding does not contribute to the overall rate , the apparent $^{18}O$-KIE is expected to reflect the intrinsic $^{18}O$-KIE of the rate-limiting $O_2$ reduction step. Accordingly, the intrinsic $^{18}O$-KIE for the rate-limiting step of $O_2^{\bullet-}$ or $FLOO^-$ formation is likely between 1.030 and 1.035, based on both calculated $^{18}O$-EIEs for these reactions (1.033-1.034) (Roth and Klinman 2003), and on the maximum $^{18}O$-KIEs observed for glucose oxidase (1.0341 ± 0.0005) and KMO (1.0304 ± 0.0003). The lower $^{18}O$-KIEs (1.019-1.0.23), particularly for cholesterol, choline, and glycolate oxidase, can thus still arise from a rate-limiting $O_2^{\bullet-}$ or $FLOO^-$ formation, but with increasing contributions from a relatively slower $O_2$ binding to the overall reaction rate that is likely associated with an intrinsic isotope effect close to unity because, upon binding, no bond changes occur in $O_2$."

"**Appendix D: Derivation of apparent correlation between $^{18}O$-KIE and $K_m(O_2)$**

To reconcile the apparent correlation between $^{18}O$-KIEs and $K_m(O_2)$ as shown in Fig. 4, we consider a simple two-step enzymatic reaction involving a reversible $O_2$ binding step and an irreversible reaction step converting enzyme-bound $O_2$ into products (either $H_2O_2$ or hydroxylated organic substrate) as shown in Eq. D1,

$$O_2 \overset{k_1}{\underset{k_2}{\leftrightarrow}} E\text{-}O_2 \overset{k_3}{\to} P \tag{D1}$$

where $E$-$O_2$ is the enzyme-bound $O_2$, P represents the reaction products, and $k_1$, $k_2$, and $k_3$ are elementary reaction rate constants of the forward and backward reactions. In this case, the measured $^{18}O$-KIE is related to the intrinsic equilibrium and kinetic isotope effects of the two elementary steps through the forward commitment to catalysis, $k_3/k_2$, as shown in Eq. D2 (Cleland 2005, Enzyme mechanisms from isotope effects in: *Isotope effects in chemistry and biology*, 915-930),

$$^{18}\text{O-KIE}_{\text{measured}} = \frac{\text{EIE}_1\text{KIE}_3 + {k_3}/{k_2}\text{KIE}_1}{1 + {k_3}/{k_2}} \tag{D2}$$

where $\text{EIE}_1$ is the equilibrium isotope effect of the $O_2$ binding step and $\text{KIE}_1$ and $\text{KIE}_3$ are the kinetic isotope effects of the $O_2$ binding and reaction steps associated with rates $k_1$ and $k_3$, respectively. From Eq. D2, two extreme cases can be derived. If $O_2$ binding alone is rate-limiting ($k_3>>k_2$), the measured $^{18}\text{O-KIE}$ will approximate $\text{KIE}_1$. If the second reaction step is rate-limiting ($k_3<<k_2$), the measured $^{18}\text{O-KIE}$ will approximate the product of $\text{EIE}_1$ and $\text{KIE}_3$. When we start with the latter case, which has a small $k_3/k_2$, and increase the forward commitment gradually, the measured $^{18}\text{O-KIE}$ will slowly decrease assuming $\text{KIE}_1$ is smaller than $\text{EIE}_1\text{KIE}_3$. For such a reaction, plotting measured $^{18}\text{O-KIEs}$ vs. $k_3/k_2$ will yield a similar (apparently linear) trend as shown in Fig. 4 as long as the commitment factor ($k_3/k_2$) is below 1. As shown in Eq. D3, $k_3/k_2$ can be related to $K_m(O_2)$, if we consider $K_m(O_2)$ to be $(k_3 + k_2)/k_1$ and $K_d$, the dissociation constant of $O_2$, to be $k_2/k_1$. The trend observed in Fig. 4 can thus be explained if Kd varies much less than $K_m(O_2)$ for this set of enzymes.

$$\frac{k_3}{k_2} = \frac{k_3}{k_2} + \frac{k_2}{k_2} - 1 = \frac{k_3+k_2}{k_2} - 1 = \frac{\text{K}_m(O_2)}{\text{K}_d} - 1 \tag{D3}$$"

**Ref 2.4** p.19. It is very interesting that there is a single example where the lambda value for comparison of $^{18}\text{O}$ to $^{17}\text{O}$ KIEs deviates from expectation. Since the two isotopes of oxygen have a different spin, this may suggest an unexpected spin component in the reaction.

**Auth 2.4** This is indeed an interesting possibility. However, there is no evidence, as far as we know, for a possible reaction step associated with the suggested reaction mechanisms of flavin-dependent enzymes that would point towards such an unexpected spin component. As we already stated in line 477, and given the breath of our current study, we retain our opinion that "this reduction mechanism cannot be further elucidated in this study". No changes were made.

**Ref 2.5** In comparing the 18O values for Cu and Fe enzymes, the authors may wish to take a look at the different 18O EIEs for model Fe and Cu dependent systems (Tian and Klinman, JACS 114, 7117 (1993).

**Auth 2.5** We have included a reference to Tian and Klinman (1993, https://doi.org/10.1021/ja00073a001) in lines 528-530, where we discuss KIEs for iron-dependent enzymes. For copper-dependent enzymes, the one additional experimental value provided in Tian and Klinman (1993) does not change the range of reported values, which we

already gathered from more recent studies. Thus, we have not included this reference in the section discussing isotope effects of copper-dependent enzymes.

**Lines 538-540:** "Calculated _or measured_ $^{18}$O-EIEs are also similar in magnitude, with 1.004-1.009 for iron-superoxo formation, 1.011-1.017 for iron-hydroperoxo formation, and 1.029 for iron-oxo formation (_Tian and Klinman 1993;_ Mirica et al. 2008)."